# Tester-Learners for Halfspaces: Universal Algorithms

**Aravind Gollakota**
Apple
`aravindg@cs.utexas.edu`

**Adam R. Klivans**
UT Austin
`klivans@cs.utexas.edu`

**Konstantinos Stavropoulos**
UT Austin
`kstavrop@cs.utexas.edu`

**Arsen Vasilyan**
MIT
`vasilyan@mit.edu`

## Abstract

We give the first tester-learner for halfspaces that succeeds *universally* over a wide class of structured distributions. Our universal tester-learner runs in fully polynomial time and has the following guarantee: the learner achieves error $O(\mathsf{opt}) + \epsilon$ on any labeled distribution that the tester accepts, and moreover, the tester accepts whenever the marginal is *any* distribution that satisfies a Poincaré inequality. In contrast to prior work on testable learning, our tester is not tailored to any single target distribution but rather succeeds for an entire target class of distributions. The class of Poincaré distributions includes all strongly log-concave distributions, and, assuming the Kannan–Lóvasz–Simonovits (KLS) conjecture, includes all log-concave distributions. In the special case where the label noise is known to be Massart, our tester-learner achieves error $\mathsf{opt} + \epsilon$ while accepting all log-concave distributions unconditionally (without assuming KLS). Our tests rely on checking hypercontractivity of the unknown distribution using a sum-of-squares (SOS) program, and crucially make use of the fact that Poincaré distributions are certifiably hypercontractive in the SOS framework.

## 1   Introduction

In this paper we study the recent model of testable learning, due to Rubinfeld and Vasilyan [RV23]. Testable learning addresses a key issue with essentially all known algorithms for the basic problem of agnostic learning, in which a learner is required to produce a hypothesis competitive with the best-fitting hypothesis in a concept class $\mathcal{C}$. The issue is that these algorithms make distributional assumptions (such as Gaussianity or log-concavity) that are in general hard to verify. This means that in the absence of any prior information about the distribution or the optimal achievable error, it can be hard to check that the learner has even succeeded at meeting its guarantee.

In the testable learning model, the learning algorithm, or tester-learner, is given access to labeled examples from an unknown distribution and may either reject or accept the unknown distribution. If it accepts, it must successfully produce a near-optimal hypothesis. Moreover, it is also required to accept whenever the unknown distribution truly has a certain target marginal $D^*$. Work of [RV23, GKK23, GKSV23, DKK+23] provided tester-learners for a range of basic classes (including halfspaces, intersections of halfspaces, and more) with respect to particular target marginals $D^*$ (such as the standard Gaussian). All of these algorithms, however, have the shortcoming that they are closely tailored to the particular target marginal $D^*$ that is chosen. Indeed, their tests would reject many well-behaved distributions that are appreciably far from $D^*$. A highly natural question from both a theoretical and a practical perspective is: can we design tester-learners that accept a wide class of distributions simultaneously, without being tailored to any particular one?

37th Conference on Neural Information Processing Systems (NeurIPS 2023).

In this work we answer this question in the affirmative by introducing and studying *universally testable learning*. We formally define this model as follows.

**Definition 1.1** (Universally Testable Learning). Let $\mathcal{C}$ be a concept class mapping $\mathbb{R}^d$ to $\{\pm 1\}$. Let $\mathcal{D}$ be a family of distributions over $\mathbb{R}^d$. Let $\epsilon, \delta > 0$ be parameters, and let $\psi : [0,1] \to [0,1]$ be some function. We say $\mathcal{C}$ can be universally testably learned w.r.t. $\mathcal{D}$ up to error $\psi(\mathsf{opt}) + \epsilon$ with failure probability $\delta$ if there exists a tester-learner $A$ meeting the following specification. For any distribution $D_{\mathcal{X}\mathcal{Y}}$ on $\mathbb{R}^d \times \{\pm 1\}$, $A$ takes in a large sample $S$ drawn from $D_{\mathcal{X}\mathcal{Y}}$, and either rejects $S$ or accepts and produces a hypothesis $h : \mathbb{R}^d \to \{\pm 1\}$ such that:

(a) (Soundness.) With probability at least $1 - \delta$ over the sample $S$ the following is true:

If $A$ accepts, then the output $h$ satisfies $\mathbb{P}_{(\mathbf{x},y)\sim D_{\mathcal{X}\mathcal{Y}}}[h(\mathbf{x}) \neq y] \leq \psi(\mathsf{opt}(\mathcal{C}, D_{\mathcal{X}\mathcal{Y}})) + \epsilon$, where $\mathsf{opt}(\mathcal{C}, D_{\mathcal{X}\mathcal{Y}}) = \inf_{f \in \mathcal{C}} \mathbb{P}_{(\mathbf{x},y)\sim D_{\mathcal{X}\mathcal{Y}}}[h(\mathbf{x}) \neq y]$.

(b) (Completeness.) Whenever the marginal of $D_{\mathcal{X}\mathcal{Y}}$ lies within $\mathcal{D}$, $A$ accepts with probability at least $1 - \delta$ over the sample $S$.

In this terminology, the original definition of testable learning reduces to the special case where $\mathcal{D} = \{D^*\}$. We stress that while the prior work of [GKK23] allowed $D^*$ to be, say, any fixed strongly log-concave distribution, their tester-learners are still tailored to the particular $D^*$ that is selected. This is because their tests rely on checking that the unknown distribution closely matches moments with $D^*$. By contrast, a universal tester-learner must accept *all* marginals in a family $\mathcal{D}$.

Our main contribution in this paper is the first universal tester-learner for the class of halfspaces with respect to a broad family of structured continuous distributions. This family is the set of all distributions with bounded Poincaré constant (see Definition 2.4) and some mild concentration and anti-concentration properties (see Definition 2.1). It captures all strongly log-concave distributions, and in fact, under the well-known Kannan–Lóvasz–Simonovits (KLS) conjecture (see Conjecture 2.6), it captures all log-concave distributions as well. Our universal tester-learner significantly generalizes the main result of [GKSV23], who showed comparable guarantees only for the case where the target marginal is the standard Gaussian.

**Theorem 1.2** (Universal Tester-Learner for Halfspaces; formally stated as Theorem 4.1). *Let $\mathcal{C}$ be the class of origin-centered halfspaces over $\mathbb{R}^d$. Let $\mathcal{D}$ be the class of $\Theta(1)$-nice and $\Theta(1)$-Poincaré distributions (see Definitions 2.1 and 2.4), which includes all isotropic strongly log-concave and, under KLS, all isotropic log-concave distributions. Then $\mathcal{C}$ can be universally testably learned w.r.t. $\mathcal{D}$ up to error $O(\mathsf{opt}) + \epsilon$ in $\mathrm{poly}(d, \frac{1}{\epsilon})$ time and sample complexity.*

A special and well-studied case of interest is when the label noise follows the Massart model, i.e. the label of every example is flipped by an adversary with probability at most $\eta$. In this case we are able to handle a considerably larger class $\mathcal{D}$ while also providing a stronger guarantee.

**Theorem 1.3** (Universal Tester-Learner for Massart Halfspaces; formally stated as Theorem 4.1). *Let $\mathcal{C}$ be the class of origin-centered halfspaces over $\mathbb{R}^d$. Let $\mathcal{D}$ be the class of $\mathrm{poly}(d)$-nice and $\mathrm{poly}(d)$-Poincaré distributions, which includes all isotropic log-concave distributions (unconditionally). Suppose the label noise follows the Massart model with noise rate at most $\eta < \frac{1}{2}$. Then $\mathcal{C}$ can be universally testably learned w.r.t. $\mathcal{D}$ up to error $\mathsf{opt} + \epsilon$ in $\mathrm{poly}(d, \frac{1}{\epsilon}, \frac{1}{1-2\eta})$ time and sample complexity.*

**Technical Overview.** We first describe the key reasons why prior tester-learners were tailored to a specific target $D^*$. All known polynomial-time algorithms for agnostically learning halfspaces up to error $O(\mathsf{opt}) + \epsilon$ require some concentration and anti-concentration properties from the input marginal distribution (encapsulated e.g. in Definition 2.1). While concentration is relatively straightforward to check (e.g. by checking that the moments do not grow at too fast a rate), the key challenge in designing tester-learners for halfspaces is to check anti-concentration. All prior tester-learners [RV23, GKK23, GKSV23, DKK$^+$23] use the heavy machinery of moment-matching to achieve this. This approach relies on establishing structural properties of the following type: if $D^*$ is a well-behaved distribution (e.g. a strongly log-concave distribution), and $D$ approximately matches $D^*$ in its low-degree moments, then $D$ is also well-behaved (in particular, anti-concentrated). A canonical statement of such a property is the main pseudorandomness result of [GKK23] (see Theorem 5.6 therein), which establishes that approximate moment-matching fools functions of a constant number of halfspaces. Applying this property inherently requires comparing the low-degree moments of

$D$ with those of $D^*$. Such tests do (implicitly) succeed universally for the class of all distributions that match low-degree moments with $D^*$ (e.g., if $D^*$ is the uniform distribution over the hypercube, moment matching would accept all $k$-wise independent distributions). Definition 1.1, however, seeks a far broader kind of universality. Our tests are not tailored to a single target in any way, and are intended to succeed over practical classes of distributions that are commonly considered in learning theory (e.g., log-concave distributions).[1]

We overcome this hurdle and design a conceptually simple way of checking anti-concentration without requiring the hammer of moment-matching. Our approach follows and improves on the roadmap used by [GKSV23] to design efficient tester-learners for halfspaces using non-convex SGD (building on [DKTZ20a, DKTZ20b]). Let us outline this approach at a high level (a more detailed technical overview may be found in [GKSV23, Sec 1.2]). The tester-learner first computes a stationary point $\mathbf{w}$ of a certain smooth version of the ramp loss, a surrogate for the 0-1 loss. Let $\mathbf{w}^*$ be any solution achieving 0-1 error opt. The tester-learner now checks distributional properties of the unknown marginal $D$ that ensure that $\mathbf{w}$ is close in angular distance to $\mathbf{w}^*$ (specifically, they ensure the contrapositive, namely that any $\mathbf{w}$ that has large gradient norm must have large angle with $\mathbf{w}^*$). By a more careful analysis of the gradient norm than in [GKSV23] (see Proposition 4.2), we are able to reduce to showing the following weak anti-concentration property. Let $\mathbf{v}$ denote any unit vector orthogonal to $\mathbf{w}$, and let $D_T$ denote $D$ restricted to the band $T = \{\mathbf{x} \mid |\langle \mathbf{w}, \mathbf{x} \rangle| \leq \sigma\}$ (where the width $\sigma$ is carefully selected according to certain constraints). Then the property we need is that

$$\mathbb{P}_{\mathbf{x} \sim D_T}[|\langle \mathbf{v}, \mathbf{x} \rangle| \geq \Theta(1)] \geq \Theta(1).$$

Our key observation is that the classical Paley–Zygmund inequality applied to the random variable $Z = \langle \mathbf{v}, \mathbf{x} \rangle^2$, where $\mathbf{x} \sim D_T$, already gives us the following type of anti-concentration:

$$\mathbb{P}\left[Z > \frac{\mathbb{E}[Z]}{2}\right] \geq \frac{1}{4} \cdot \frac{\mathbb{E}[Z]^2}{\mathbb{E}[Z^2]}.$$

This turns out to suffice for our purposes—provided we can show a hypercontractivity property for $Z$, namely that $\mathbb{E}[Z^2] \leq \Theta(1) \mathbb{E}[Z]^2$ (as well as that $\mathbb{E}[Z] = \Theta(1)$, which is just a second moment constraint).

Our main algorithmic idea is to use a sum-of-squares (SOS) program to check hypercontractivity of the random variable $Z$. To do so, we crucially leverage a result due to [KS17] stating that any $D$ that has bounded Poincaré constant is *certifiably hypercontractive* in the SOS framework (and it turns out this extends to $D_T$ as well). This means that we can run a certain polynomial-time semidefinite program that checks hypercontractivity of $Z$ over the sample, and whenever $D$ is in fact Poincaré, we are guaranteed that the test will pass with high probability (see Proposition 3.5). This is sufficient to ensure that the stationary point $\mathbf{w}$ we have computed is indeed close in angular distance to $\mathbf{w}^*$.

In order to finally arrive at our main results, we need to run further tests which ensure that the disagreement between our computed $\mathbf{w}$ and any (unknown) optimum $\mathbf{w}^*$ is bounded by the angle between them, i.e., $\mathbb{P}_{\mathbf{x} \sim D}[\text{sign}(\langle \mathbf{w}, \mathbf{x} \rangle \neq \text{sign}(\langle \mathbf{w}^*, \mathbf{x} \rangle)] \leq O(\angle(\mathbf{w}, \mathbf{w}^*))$ (see Lemma 3.1). This in turn guarantees that $\mathbf{w}$ has error $O(\text{opt}) + \epsilon$. We stress that while [GKSV23] introduced similar testers for the special case of Gaussian marginals, our tests succeed universally with respect to a broad family of distributions including some heavy-tailed distributions (see Definition 2.1). From a technical perspective, prior to our work, such tests either produced a suboptimal bound, or required estimating the operator norms of a polynomial number of random matrices formed using rejection sampling. We significantly simplify this approach by showing that it is sufficient to estimate the operator norm of a single random matrix. Finally, to obtain our improved results for the Massart setting, it turns out that the proof admits certain simplifications that guarantee final error $\text{opt} + \epsilon$ while also allowing a wider range of Poincaré distributions.

**Related Work.** There is a large body of work on agnostic learning algorithms for halfspaces that run in fully polynomial time. We briefly mention only those that are most closely relevant to our work; please see [BH21] for a survey as well as [GKSV23, Sec 1.1] for further related work. Following a

---

[1]One may wonder if it is possible to test whether the low-degree moments of the input marginal $D$ match *any* distribution in a family $\mathcal{D}$ (e.g., all strongly log-concave distributions) without directly comparing to a specific $D^*$. This is a reduction to testing whether a given (approximate) low-degree moment tensor lies within a large set of target low-degree moment tensors, and would indeed suffice for universally testable learning. This general problem, however, seems highly challenging to solve directly.

long line of work on distribution-specific agnostic learners for halfspaces [KLS09, ABL17, Dan15, BZ17, YZ17, Zha18, ZSA20, ZL21], the work of [DKTZ20a] introduced a particularly simple approach for the Massart setting, based solely on non-convex SGD. This work, which sets the template that our approach also follows, achieved the information-theoretically optimal error of $\mathsf{opt} + \epsilon$ for origin-centered Massart halfspaces over a wide range of structured distributions (and was later extended to general halfspaces by [DKK$^+$22]). The non-convex SGD approach was then generalized by [DKTZ20b] to show an $O(\mathsf{opt}) + \epsilon$ guarantee for the fully agnostic setting.

The testable learning model was introduced by the work of [RV23], who showed a tester-learner for halfspaces achieving error $\mathsf{opt} + \epsilon$ in time $d^{\widetilde{O}(1/\epsilon^4)}$ for the case where the target marginal is Gaussian. Subsequently, [GKK23] provided a general algorithmic framework based on moment-matching for this problem, and showed a tester-learner for halfspaces only requiring time $d^{\widetilde{O}(1/\epsilon^2)}$ with respect to any fixed strongly log-concave marginal (matching known lower bounds for ordinary agnostic learning over Gaussian marginals [GGK20, DKZ20, DKPZ21, DKR23]).

The most closely relevant work to the present one is that of [GKSV23] (see also [DKK$^+$23]), who showed fully polynomial-time tester-learners for halfspaces achieving error $O(\mathsf{opt}) + \epsilon$ in the agnostic setting and $\mathsf{opt} + \epsilon$ in the Massart setting for the case where the target marginal is the Gaussian. As detailed in the technical overview, their tests rely crucially on moment-matching and are tailored to a specific target marginal. By contrast, our tests check hypercontractivity using an SOS program and succeed universally for a wide class of certifiably hypercontractive distributions.

Certifying distributional properties such as hypercontractivity is an important aspect of a large body of work on robust algorithmic statistics using the SOS framework. We will not attempt to summarize this literature here and direct the reader to [KS17, BK21] for overviews of related work, as well as to [FKP$^+$19] for a textbook treatment. The notion of certifiable anti-concentration has also been studied (see e.g. [KKK19, RY20, BK21]), but it turns out not to be directly useful for our purposes as it is only known to hold for distributions satisfying very strong conditions such as rotational symmetry.

**Limitations and Further Work.** Open directions in testable learning (and universally testable learning) include the design of (efficient) tester-learners for concept classes other than the class of halfspaces, e.g., functions of halfspaces or neurons with other activations (like ReLU or sigmoid).

## 2 Preliminaries

**Notation and Terminology.** For what follows, we consider $D_{\mathcal{XY}}$ to be an unknown joint distribution over $\mathcal{X} \times \mathcal{Y}$ from which we receive independent samples, and its marginal on $\mathcal{X}$ will be denoted by $D_{\mathcal{X}}$. In particular $\mathcal{X} = \mathbb{R}^d$, and labels will lie in $\mathcal{Y} = \{\pm 1\}$. We will use $\mathcal{C}$ to denote a concept class mapping $\mathbb{R}^d$ to $\{\pm 1\}$, which throughout this paper will be the class of halfspaces or functions of halfspaces over $\mathbb{R}^d$. We use $\mathsf{opt}(\mathcal{C}, D_{\mathcal{XY}})$ to denote the optimal error $\inf_{f \in \mathcal{C}} \mathbb{P}_{(\mathbf{x},y) \sim D_{\mathcal{XY}}}[f(\mathbf{x}) \neq y]$, or just $\mathsf{opt}$ when $\mathcal{C}$ and $D_{\mathcal{XY}}$ are clear from context. We recall that in Massart noise model, the labels satisfy $\mathbb{P}_{y \sim D_{\mathcal{XY}}|\mathbf{x}}[y \neq \mathrm{sign}(\langle \mathbf{w}^*, \mathbf{x} \rangle) \mid \mathbf{x}] = \eta(\mathbf{x})$, with $\eta(\mathbf{x}) \leq \eta < \frac{1}{2}$ for all $\mathbf{x}$. When we have adversarial noise (i.e., when we are in the agnostic model), the labels can be completely arbitrary. In both cases, the goal is to produce a hypothesis whose error is competitive with $\mathsf{opt}$. We use $\mathbb{E}$ to denote the expectation of a random variable in brackets (or, correspondingly, $\mathbb{P}$ for the probability of an event), either over the unknown joint distribution or over the empirical distribution with respect to a sample $S$ (e.g., $\mathbb{E}_{Z \in S}[f(Z)] = \frac{1}{|S|} \sum_{Z \in S} f(Z)$).

**Definitions and Distributional Assumptions.** For the problem of learning halfspaces in the agnostic and in Massart noise models, any of the known polynomial algorithms that achieve computationally optimal guarantees require that the marginal distribution has at least the following nice properties previously defined by, e.g., [DKTZ20b].

**Definition 2.1** (Nice Distributions). For a given constant $\lambda \geq 1$, we consider the class of $\lambda$-nice distributions over $\mathbb{R}^d$ to be the distributions that satisfy the following properties:

1. For any unit vector $\mathbf{v}$ in $\mathbb{R}^d$ the distribution satisfies $\mathbb{E}[\langle \mathbf{v}, \mathbf{x} \rangle^2] \in [\frac{1}{\lambda}, \lambda]$.(bounded spectrum)

2. For any two dimensional subspace $V$, the corresponding marginal density $q_V(\mathbf{x})$ satisfies $q_V(\mathbf{x}) \geq 1/\lambda$ for any $\|\mathbf{x}\|_2 \leq 1/\lambda$. (anti-anti-concentration)

3. For any two dimensional subspace $V$, the corresponding marginal density $q_V(\mathbf{x})$ satisfies $q_V(\mathbf{x}) \le Q(\|\mathbf{x}\|_2)$ for some function $Q : \mathbb{R}_+ \to \mathbb{R}_+$ such that $\sup_{r \ge 0} Q(r) \le \lambda$ and also $\int_{r=0}^{\infty} r^k Q(r)\, dr \le \lambda$, for any $k = 1, 3, 5$.      (anti-concentration and concentration)

In the testable learning framework, however, corresponding results provide testable guarantees with respect to target marginals that are isotropic strongly log-concave [GKSV23], which is a strictly stronger condition than the one of Definition 2.1 (see Proposition 2.3 below). We now provide the standard definition of (strongly) log-concave distributions.

**Definition 2.2** ((Strongly) Log-Concave Distributions [SW14]). We say that a distribution over $\mathbb{R}^d$ is ($\beta$-strongly) log-concave, if its density can be written as $e^{-\varphi}$, where $\varphi$ is a ($\beta$-strongly) convex function on $\mathbb{R}^d$ (for some $\beta > 0$).

**Proposition 2.3** (Log-Concave Distributions are Nice [LV07]). *There exists a universal constant $\lambda \ge 1$ such that any isotropic log-concave distribution is $\lambda$-nice.*

In this work, we provide universally testable guarantees with respect to the class of nice distributions with bounded Poincaré constant (see Definition 2.4 below).

**Definition 2.4** (Poincaré Distributions). For a given value $\gamma > 0$, we say that a distribution over $\mathbb{R}^d$ is $\gamma$-Poincaré, if $\mathrm{Var}(f(\mathbf{x})) \le \gamma \cdot \mathbb{E}[\|\nabla f(\mathbf{x})\|_2^2]$ for any differentiable function $f : \mathbb{R}^d \to \mathbb{R}$.

Although it is not clear whether one can efficiently obtain testable guarantees for the problem of learning noisy halfspaces under nice marginals (which is known to be an efficiently solvable problem in the non-testable setting [DKTZ20a, DKTZ20b]), by restricting our attention to nice distributions that, additionally, have bounded Poincaré constant, we obtain efficient learning results, even in the universally testable setting. Our results are strictly stronger than the ones in [GKSV23], since we capture isotropic strongly log-concave distributions universally, due to Proposition 2.3 and the fact that strongly log-concave distributions are also Poincaré, as per Proposition 2.5 below.

**Proposition 2.5** (Strongly Log-Concave Distributions are Poincaré, [SW14, Proposition 10.1]). *Any $\frac{1}{\gamma}$-strongly log-concave distribution is $\gamma$-Poincaré.*

Furthermore, under a long-standing conjecture about the geometry of convex bodies [KLS95], our results capture the family of all isotropic log-concave distributions.

**Conjecture 2.6** (Kannan–Lovász–Simonovits Conjecture [KLS95] reformulation from [LV18]). *There is a universal constant $\gamma > 0$ for which any isotropic log-concave distribution is $\gamma$-Poincaré.*

## 3 Universal Testers

In this section, we present two basic testers that constitute the basic building blocks of the universal tester-learners we provide in the next section. The testers in this section might be of independent interest and their appeal is that they succeed even when the distribution in their input is unspecified up to certain bounds on a number of its statistics. In fact, the family of distributions for which each such tester succeeds is of infinite size, even non-parametric.

### 3.1 Universal Tester for Bounding Local Halfspace Disagreement

First, we present a universal tester that checks, given a parameter vector $\mathbf{w}$, whether a set of samples $S$ is such that bounding the angular distance of $\mathbf{w}$ from an optimum parameter vector, implies that the corresponding halfspace disagrees with the (unknown) optimum halfspace only on a bounded fraction of points in $S$. This property ensures that if $\mathbf{w}$ is close to the optimum parameter vector, then it is also an approximate empirical risk minimizer. The tester universally accepts samples from nice distributions with high probability (Definition 2.1).

**Lemma 3.1** (Universally Testable Bound for Local Halfspace Disagreement). *Let $D_{\mathcal{X}\mathcal{Y}}$ be a distribution over $\mathbb{R}^d \times \{\pm 1\}$, $\mathbf{w} \in \mathbb{S}^{d-1}$, $\theta \in (0, \pi/4]$, $\lambda \ge 1$ and $\delta \in (0, 1)$. Then, for a sufficiently large constant $C$, there is a tester that given $\delta$, $\theta$, $\mathbf{w}$ and a set $S$ of samples from $D_{\mathcal{X}}$ with size at least $C \cdot \left(\frac{d^4}{\theta^2 \delta}\right)$, runs in time $\mathrm{poly}\left(d, \frac{1}{\theta}, \frac{1}{\delta}\right)$ and satisfies the following specifications:*

    *(a) If the tester accepts $S$, then for every unit vector $\mathbf{w}' \in \mathbb{R}^n$ satisfying $\angle(\mathbf{w}, \mathbf{w}') \le \theta$ we have*

$$\mathbb{P}_{\mathbf{x} \sim S}[\mathrm{sign}(\langle \mathbf{w}', \mathbf{x} \rangle) \ne \mathrm{sign}(\langle \mathbf{w}, \mathbf{x} \rangle)] \le C \cdot \theta \cdot \lambda^C$$

(b) *If the distribution $D_{\mathcal{X}}$ is $\lambda$-nice, the tester accepts $S$ with probability $1 - \delta$.*

The proof of Lemma 3.1 simplifies and improves the proof of a similar but weaker result in [GKSV23] (see their Proposition 4.5). The initial tester exploited the observation that the probability of disagreement between two halfspaces can be upper bounded by a sum of products, where each product has two terms: one corresponding to the probability of falling in a (known) strip orthogonal to $\mathbf{w}$ and one corresponding to the probability of having large enough inner product with some unknown vector orthogonal to $\mathbf{w}$, conditioned in the (known) strip. The first term can be controlled by estimating the probability of falling in a (known) strip, while the second follows by Chebyshev's inequality, after estimating the largest eigenvalue of the covariance matrix conditioned in the known strip. This approach introduces a number of complications, including the fact that conditioning requires rejection sampling, which, in turn requires a lower bound on the probability of falling inside each strip. We propose a simpler tester that controls all of the terms of the sum simultaneously by estimating the largest eigenvalue of a single covariance matrix (without conditioning). Upper and lower bounds on the eigenvalues of random symmetric matrices can be universally tested with testers that are guaranteed to accept when the elements of the matrix have bounded second moments (spectral tester of Proposition A.2). We present our full proof in Appendix B.1.

## 3.2 Universally Testable Weak Anti-Concentration

We now provide an important universal tester, which ensures that for a given vector $\mathbf{w}$, a sample set $S$ and any unknown unit vector $\mathbf{v}$ orthogonal to $\mathbf{w}$, among the samples falling within a (known) strip orthogonal to $\mathbf{w}$, at least a constant fraction is absolutely correlated with $\mathbf{v}$ by a constant. In other words, the tester ensures that the conditional empirical distribution is weakly anti-concentrated in every direction. The tester universally accepts nice distributions that have bounded Poincaré constant.

**Lemma 3.2** (Universally Testable Weak Anti-Concentration). *Let $D$ be a distribution over $\mathbb{R}^d$. Then, there is a universal constant $C > 0$ and a tester that given a unit vector $\mathbf{w} \in \mathbb{R}^d$, $\delta \in (0, 1)$, $\gamma > 0$, $\lambda \geq 1$, $\sigma \leq \frac{1}{2\lambda}$ and a set $S$ of i.i.d. samples from $D$ with size at least $C \cdot \frac{d^4}{\sigma^2 \delta} \log(d) \lambda^C$, runs in time $\mathrm{poly}(d, \lambda, \frac{1}{\sigma}, \frac{1}{\delta}, \log(\frac{1}{\gamma}))$ and satisfies the following specifications*

(a) *If the tester accepts $S$, then for any unit vector $\mathbf{v} \in \mathbb{R}^d$ with $\langle \mathbf{v}, \mathbf{w} \rangle = 0$ we have*

$$\mathbb{P}_{\mathbf{x} \in S}\left[ |\langle \mathbf{v}, \mathbf{x} \rangle| \geq \frac{1}{C\lambda^C} \; \Big| \; |\langle \mathbf{w}, \mathbf{x} \rangle| \leq \sigma \right] \geq \frac{1}{C\lambda^C \gamma^4}$$

(b) *If $D$ is $\gamma$-Poincaré and $\lambda$-nice, then the tester accepts $S$ with probability at least $1 - \delta$.*

The proof of Lemma 3.2 is based on a simple fact from probability that is true for any non-negative random variable and ensures that the mass assigned to the tails is lower bounded by the ratio of the square of its expectation to the second moment.

**Proposition 3.3** (Paley–Zygmund Inequality). *For any non-negative random variable $Z$, we have*

$$\mathbb{P}[Z > \mathbb{E}[Z]/2] \geq \frac{1}{4} \cdot \frac{\mathbb{E}[Z]^2}{\mathbb{E}[Z^2]}$$

In the special case where $Z$ follows the distribution of $\langle \mathbf{v}, \mathbf{x} \rangle^2$ conditioned on $|\langle \mathbf{w}, \mathbf{x} \rangle| \leq \sigma$ for some unitary orthogonal vectors $\mathbf{v}, \mathbf{w}$, some $\sigma > 0$ and some random variable $\mathbf{x}$ whose distribution is, say, 1-nice (see Definition 2.1), one can show that $\mathbb{E}[Z]$ is lower bounded by a constant and $\mathbb{E}[Z^2]$ is upper bounded by another constant, so $Z$ assigns a non-trivial mass to a set that is bounded away from zero. This property is useful in the context of learning noisy halfspaces, as we show in the following section (see Proposition 4.2 and Lemma 4.3). However, testing algorithms that check whether such a property holds for given $\mathbf{w}$ and $\sigma$, are guaranteed to succeed when the marginal distribution has, additionally, bounded Poincaré constant. The main part of the proof that requires a bounded Poincaré constant, is testing whether $\mathbb{E}[Z^2]$ is bounded uniformly over the set of unit vectors $\mathbf{v}$ orthogonal to $\mathbf{w}$, since $Z^2 = \langle \mathbf{v}, \mathbf{x} \rangle^4$, where $\mathbf{v}$ is unknown. We use the following result from [KS17].

**Proposition 3.4** (Certifiable Hypercontractivity of Poincaré Distributions, Theorem 4.1 in [KS17]). *Let $\delta \in (0, 1)$, $\gamma > 0$ and let $D$ be a $\gamma$-Poincaré distribution over $\mathbb{R}^d$. Let $S$ be a set of independent samples from $D$ with size at least $(2d \log(4d/\delta))^4$. Consider the constrained maximization problem*

$$\arg \max_{\|\mathbf{v}\|_2 = 1} \mathbb{E}_{\mathbf{x} \in S}[\langle \mathbf{v}, \mathbf{x} \rangle^4] \tag{3.1}$$

*Then, the optimum solution of the degree-4 sum-of-squares relaxation of the problem* (3.1) *has value at most $C\gamma^4$ for some universal constant $C$, with probability at least $1 - \delta$ over the sample $S$.*

Using Proposition 3.4, we are able to provide a universal tester for bounding the empirical fourth moments. The tester solves an appropriate SDP relaxation of the (hard) problem [HL13] of finding the direction with maximum fourth moment and is guaranteed to succeed if $\mathbf{x}$ has Poincaré parameter bounded by a known value.

**Proposition 3.5** (Hypercontractivity Tester). *Let $D$ be a distribution over $\mathbb{R}^d$. Then, there is a tester that given $\delta \in (0, 1)$, $\gamma > 0$ and a set $S$ of i.i.d. samples from $D$ with size at least $(2d \log(4d/\delta))^4$, runs in time $\mathrm{poly}(d, \log \frac{1}{\delta}, \log \frac{1}{\gamma})$ and satisfies the following specifications*

(a) *If the tester accepts $S$, then for any unit vector $\mathbf{v} \in \mathbb{R}^d$ we have*

$$\mathbb{E}_{\mathbf{x} \in S}[\langle \mathbf{v}, \mathbf{x} \rangle^4] \leq C \cdot \gamma^4 \,, \text{ where $C$ is some universal constant.}$$

(b) *If the distribution $D$ is $\gamma$-Poincaré, then the tester accepts $S$ with probability at least $1 - \delta$.*

*Proof.* The tester does the following:

1. Solves a degree-4 sum-of-squares relaxation of problem (3.1) up to accuracy $\gamma^4$. (For a formal definition of the relaxed problem, see Problem (2.3) in [KS17].)

2. If the solution has value larger than $(C - 1)\gamma^4$, then **reject**. Otherwise **accept**.

The computational complexity of the tester is $\mathrm{poly}(|S|, d, \log \frac{1}{\gamma})$, since the problem it solves can be written as a semidefinite program [Sho87, Par00, Nes00, Las01].

If the tester accepts $S$, then we know that the optimal solution of the relaxed problem is at most $C\gamma^4$ and we also know that any solution of the initial problem (3.1) has value at most equal to the value of the relaxation. Therefore $\mathbb{E}[\langle \mathbf{v}, \mathbf{x} \rangle^4] \leq C\gamma^4$, for any $\mathbf{v} \in \mathbb{S}^{d-1}$.

On the other hand, if the true distribution $D$ is $\gamma$-Poincaré, then, with probability at least $1 - \delta$, we have that the solution found in step 3.2 has, with probability at least $1 - \delta$, value at most $C'\gamma^4$ for some universal constant $C'$, due to Proposition 3.4. In order to ensure that the tester will accept with probability at least $1 - \delta$, it suffices to pick $C = C' + 1$. $\qquad\square$

We provide the full proof of Lemma 3.2, in Appendix B.2. The tests we perform include a spectral tester that accepts with high probability when the distribution of $\mathbf{x}$ is nice (similar to the spectral tester used for Lemma 3.1), a tester of the probability that $|\langle \mathbf{w}, \mathbf{x} \rangle| \leq \sigma$ and the hypercontractivity tester of Proposition 3.5.

## 4 Universal Tester-Learners for Halfspaces

In this section, we present our main result on universally testable learning of halfspaces.

**Theorem 4.1** (Efficient Universal Tester-Learner for Halfspaces). *Let $D_{\mathcal{XY}}$ be any distribution over $\mathbb{R}^d \times \{\pm 1\}$. Let $\mathcal{C}$ be the class of origin centered halfspaces in $\mathbb{R}^d$. Then, for any $\lambda \geq 1$, $\gamma > 0$, $\epsilon > 0$ and $\delta \in (0, 1)$, there exists an universal tester-learner for $\mathcal{C}$ w.r.t. the class of $\lambda$-nice and $\gamma$-Poincaré marginals up to error $\mathrm{poly}(\lambda) \cdot (1 + \gamma^4) \cdot \mathrm{opt} + \epsilon$, where $\mathrm{opt} = \min_{\mathbf{w} \in \mathbb{S}^{d-1}} \mathbb{P}_{D_{\mathcal{XY}}}[y \neq \mathrm{sign}(\langle \mathbf{w}, \mathbf{x} \rangle)]$, and error probability at most $\delta$, using a number of samples and running time $\mathrm{poly}(d, \lambda, \gamma, \frac{1}{\epsilon}, \log \frac{1}{\delta})$.*

*Moreover, if the noise is Massart with given rate $\eta < 1/2$, then the algorithm achieves error $\mathrm{opt} + \epsilon$ with time and sample complexity $\mathrm{poly}(d, \lambda, \gamma, \frac{1}{\epsilon}, \frac{1}{1-2\eta}, \log \frac{1}{\delta})$.*

Our proof follows a surrogate loss minimization approach that has been used for classical learning of noisy halfspaces [DKTZ20a, DKTZ20b] as well as classical (non-universal) testable learning [GKSV23]. In particular, the algorithm runs Projected Stochastic Gradient Descent (see A.5) on a surrogate loss whose stationary points are shown to be close to optimum parameter vectors under certain distributional assumptions. In the regular testable learning setting, given a stationary point, the above property can be tested with respect to any (fixed and known) target strongly log-concave marginal as shown by [GKSV23]. For such a stationary point, more tests are used in order to ensure

bounds on local halfspace disagreement. We provide some delicate refinements of the proofs in [GKSV23] that enable us to substitute their testers with the universal testers of Section 3.

We use the following surrogate loss function which was also used in [GKSV23].

$$\mathcal{L}_\sigma(\mathbf{w}; D_{\mathcal{X}\mathcal{Y}}) = \mathbb{E}_{(\mathbf{x},y)\sim D_{\mathcal{X}\mathcal{Y}}}\left[\ell_\sigma\left(-y\frac{\langle \mathbf{w}, \mathbf{x}\rangle}{\|\mathbf{w}\|_2}\right)\right], \tag{4.1}$$

In Equation (4.1), the function $\ell_\sigma$ is a smoothed version of the step function as in Proposition A.4.

In order to analyze the properties of the stationary points of the surrogate loss, we provide the following refinement of results implicit in [GKSV23, DKTZ20a, DKTZ20b]. We show that the gradient of the surrogate loss is lower bounded by the difference between certain quantities that are controlled by the marginal distribution (see Figure 1). We stress that we do not use any assumptions for the marginal distribution in this step. Prior work included similar bounds, but the corresponding quantities were different. We need to be more precise and provide the following result, whose proof is based on two dimensional geometry and can be found in Appendix C.1.

**Proposition 4.2** (Modification from [GKSV23, DKTZ20a, DKTZ20b]). *For a distribution $D_{\mathcal{X}\mathcal{Y}}$ over $\mathbb{R}^d \times \{\pm 1\}$ let* opt *be the minimum error achieved by some origin-centered halfspace and $\mathbf{w}^* \in \mathbb{S}^{d-1}$ a corresponding vector. Consider $\mathcal{L}_\sigma$ as in Equation (4.1) for $\sigma > 0$ and let $\eta < 1/2$. Let $\mathbf{w} \in \mathbb{S}^{d-1}$ with $\angle(\mathbf{w}, \mathbf{w}^*) = \theta < \frac{\pi}{2}$ and $\mathbf{v} \in \mathrm{span}(\mathbf{w}, \mathbf{w}^*)$ such that $\langle \mathbf{v}, \mathbf{w}\rangle = 0$ and $\langle \mathbf{v}, \mathbf{w}^*\rangle < 0$. Then, for some universal constant $C > 0$ and any $\alpha \geq \frac{\sigma}{2\tan\theta}$ we have $\|\nabla_\mathbf{w}\mathcal{L}_\sigma(\mathbf{w}; D_{\mathcal{X}\mathcal{Y}})\|_2 \geq A_1 - A_2 - A_3$, where*

$$A_1 = \frac{\alpha}{C \cdot \sigma} \cdot \mathbb{P}\left[|\langle \mathbf{v}, \mathbf{x}\rangle| \geq \alpha \;\; and \;\; |\langle \mathbf{w}, \mathbf{x}\rangle| \leq \frac{\sigma}{6}\right]$$

$$A_2 = \frac{C}{\tan\theta} \cdot \mathbb{P}\left[|\langle \mathbf{w}, \mathbf{x}\rangle| \leq \frac{\sigma}{2}\right] \;\; and \;\; A_3 = \frac{C}{\sigma} \cdot \sqrt{\mathsf{opt}} \cdot \sqrt{\mathbb{E}\left[\langle \mathbf{v}, \mathbf{x}\rangle^2 \cdot \mathbb{1}_{\{|\langle \mathbf{w}, \mathbf{x}\rangle| \leq \frac{\sigma}{2}\}}\right]}$$

*Moreover, if the noise is Massart with rate $\eta$, then $\|\nabla_\mathbf{w}\mathcal{L}_\sigma(\mathbf{w}; D_{\mathcal{X}\mathcal{Y}})\|_2 \geq (1 - 2\eta)A_1 - A_2$.*

If the marginal distribution is nice, then the quantities $A_1, A_2$ and $A_3$ are such that $\sigma$ can be chosen accordingly so that stationary points of the surrogate loss (or their inverses) are close to some optimum vector (see Proposition A.3 for properties of nice distributions). We use some simple tests (e.g., estimate the probability of falling in a strip, $\mathbb{P}[|\langle \mathbf{w}, \mathbf{x}\rangle| \leq \sigma/2]$ and appropriate spectral testers) as well as our universal tester for weak anti-concentration (see 3.2) to establish bounds on quantities $A_1, A_2$ and $A_3$ which ensure that the desired property holds for a given vector $\mathbf{w}$, under no distributional assumptions. The tester in the following result universally accepts nice distributions with bounded Poincaré parameter. The formal proof can be found in Appendix C.2.

**Lemma 4.3** (Universally Testable Structure of Surrogate Loss). *Let $D_{\mathcal{X}\mathcal{Y}}$ be any distribution over $\mathbb{R}^d \times \{\pm 1\}$. Consider $\mathcal{L}_\sigma$ as in Equation (4.1). Then, there is a universal constant $C > 0$ and a tester that given a unit vector $\mathbf{w} \in \mathbb{R}^d$, $\delta \in (0, 1)$, $\eta < 1/2$, $\gamma > 0$, $\lambda \geq 1$, $\sigma \leq \frac{1}{C\lambda^C}$ and a set $S$ of i.i.d. samples from $D_{\mathcal{X}\mathcal{Y}}$ with size at least $C \cdot \frac{d^4}{\sigma^2\delta}\log(d)\lambda^C$, runs in time $\mathrm{poly}(d, \lambda, \frac{1}{\sigma}, \frac{1}{\delta}, \log(\frac{1}{\gamma}))$ and satisfies the following specifications*

(a) *If the tester accepts $S$, then, the following statements are true for the minimum error $\mathsf{opt}_S$ achieved by some origin-centered halfspace on $S$ and the optimum vector $\mathbf{w}_S^* \in \mathbb{S}^{d-1}$*

  - *If the noise is Massart with associated rate $\eta$ and $\|\nabla_\mathbf{w}\mathcal{L}_\sigma(\mathbf{w}; S)\|_2 \leq \frac{1-2\eta}{C\lambda^C\gamma^4}$ then either $\angle(\mathbf{w}, \mathbf{w}_S^*) \leq \frac{C\lambda^C(1+\gamma^4)}{1-2\eta} \cdot \sigma$ or $\angle(-\mathbf{w}, \mathbf{w}_S^*) \leq \frac{C\lambda^C(1+\gamma^4)}{1-2\eta} \cdot \sigma$.*
  - *If the noise is adversarial with $\mathsf{opt}_S \leq \frac{\sigma}{C\lambda^C}$ and $\|\nabla_\mathbf{w}\mathcal{L}_\sigma(\mathbf{w}; S)\|_2 < \frac{1}{C\lambda^C\gamma^4}$ then either $\angle(\mathbf{w}, \mathbf{w}_S^*) \leq C\lambda^C(1+\gamma^4) \cdot \sigma$ or $\angle(-\mathbf{w}, \mathbf{w}_S^*) \leq C\lambda^C(1+\gamma^4) \cdot \sigma$.*

(b) *If the marginal $D_{\mathcal{X}}$ is $\lambda$-nice and $\gamma$-Poincaré, then the tester accepts $S$ with probability at least $1 - \delta$.*

We now give the algorithm for $\delta \leftarrow 1/3$ since we can reduce the probability of failure with repetition (repeat $O(\log\frac{1}{\delta})$ times, accept if the rate of acceptance is $\Omega(1)$ and output the halfspace achieving the minimum test error among the halfspaces returned).

The algorithm receives $\lambda \geq 1$, $\gamma > 0$, $\epsilon > 0$ and $\eta \in (0, 1/2) \cup \{1\}$ (say $\eta = 1$ when we are in the agnostic case) and does the following for some appropriately large universal constants $C_1, C_2 > 0$.

1. First, initialize $E = \frac{\epsilon}{C_1 \lambda^{C_1}}$, and let $\Sigma$ be a list of real numbers and $A$ be a positive real number, where $\Sigma$ and $A$ are defined as follows. If $\eta = 1$, then $\Sigma$ is an $\frac{E}{C_1 \lambda^{C_1}}$-cover of the interval $\left[0, \frac{1}{C_1 \lambda^{C_1}}\right]$ and $A = \frac{1}{C_1 \lambda^{C_1} \gamma^4}$. Otherwise, let $\Sigma = \left\{ \frac{E \cdot (1 - 2\eta)}{C_1 \lambda^{C_1}(1 + \gamma^4)} \right\}$ and $A = \frac{1 - 2\eta}{C_1 \lambda^{C_1} \gamma^4}$.

2. Draw a set $S_1$ of $C_2 \left( \frac{\lambda d}{\gamma \epsilon} \right)^{C_2}$ i.i.d. samples from $D_{\mathcal{X}\mathcal{Y}}$ and run PSGD, as specified in Proposition A.5 with $\epsilon \leftarrow A, \delta \leftarrow \frac{\delta}{C_1}$ on the loss $\mathcal{L}_\sigma$ for each $\sigma \in \Sigma$.

3. Form a list $L$ with all the pairs of the form $(\mathbf{w}, \sigma)$ where $\mathbf{w} \in \mathbb{S}^{d-1}$ is some iterate of the PSGD subroutine performed on $\mathcal{L}_\sigma$.

4. Draw a fresh set $S_2$ of $C_2 \left( \frac{\lambda d}{\gamma \epsilon} \right)^{C_2}$ i.i.d. samples from $D_{\mathcal{X}\mathcal{Y}}$ and compute for each $(\mathbf{w}, \sigma) \in L$ the value $\|\nabla_{\mathbf{w}} \mathcal{L}_\sigma(\mathbf{w}; S_2)\|_2$. If, for some $\sigma \in \Sigma$, $\|\nabla_{\mathbf{w}} \mathcal{L}_\sigma(\mathbf{w}; S_2)\|_2 > A$ for all $(\mathbf{w}, \sigma) \in L$, then **reject**.

5. Update $L$ by keeping for each $\sigma \in \Sigma$ only one pair of the form $(\mathbf{w}, \sigma)$ for which we have $\|\nabla_{\mathbf{w}} \mathcal{L}_\sigma(\mathbf{w}; S_2)\|_2 \leq A$.

6. Run the following tests for each $(\mathbf{w}, \sigma) \in L$. (This will ensure that part (a) of Lemma 4.3 holds for each of the elements of $L$, i.e., that any stationary point of the loss $\mathcal{L}_\sigma$ that lies in $L$ is angularly close to the empirical risk minimizer[2].).

   - If $\mathbb{P}_{(\mathbf{x},y) \in S_2}[|\langle \mathbf{w}, \mathbf{x} \rangle| \leq \frac{\sigma}{6}] \leq \frac{\sigma}{C_1 \lambda^{C_1}}$ or $\mathbb{P}_{(\mathbf{x},y) \in S_2}[|\langle \mathbf{w}, \mathbf{x} \rangle| \leq \frac{\sigma}{2}] > \sigma \cdot C_1 \lambda^{C_1}$, then **reject**.

   - Compute the $(d-1) \times (d-1)$ matrices $M_{S_2}^+$ and $M_{S_2}^-$ as follows:[3]

$$ M_{S_2}^+ = \mathop{\mathbb{E}}_{(\mathbf{x},y) \in S_2} \left[ (\mathrm{proj}_{\perp \mathbf{w}} \mathbf{x})(\mathrm{proj}_{\perp \mathbf{w}} \mathbf{x})^T \cdot \mathbb{1}_{\{|\langle \mathbf{w}, \mathbf{x} \rangle| \leq \frac{\sigma}{2}\}} \right] $$

$$ M_{S_2}^- = \mathop{\mathbb{E}}_{(\mathbf{x},y) \in S_2} \left[ (\mathrm{proj}_{\perp \mathbf{w}} \mathbf{x})(\mathrm{proj}_{\perp \mathbf{w}} \mathbf{x})^T \cdot \mathbb{1}_{\{|\langle \mathbf{w}, \mathbf{x} \rangle| \leq \frac{\sigma}{6}\}} \right] $$

   - **Reject** if the maximum singular value of $M_{S_2}^+$ is greater than $\sigma \cdot C_1 \lambda^{C_1}$.

   - **Reject** if the minimum singular value of $M_{S_2}^-$ is less than $\frac{\sigma}{C_1 \lambda^{C_1}}$.

   - Run the hypercontractivity tester on $S' = \{\mathrm{proj}_{\perp \mathbf{w}} \mathbf{x} : (\mathbf{x}, y) \in S_2 \text{ and } |\langle \mathbf{w}, \mathbf{x} \rangle| \leq \sigma\}$, i.e., solve an appropriate SDP (see Prop. 3.5 with $\gamma \leftarrow \gamma, \delta \leftarrow \delta/C_1$) and **reject** if the solution is larger than a specified threshold.

7. Set $\theta = \frac{(1 + \gamma^4)\sigma}{A\gamma^4}$, and run the following tests for each pair of the form $(\mathbf{w}, \sigma)$ and $(-\mathbf{w}, \sigma)$ where $(\mathbf{w}, \sigma) \in L$. (This will ensure that part (a) of Lemma 3.1 is activated, i.e., that the distance of a vector from the empirical risk minimizer is an accurate proxy for the error of the corresponding halfspace.)

   - If $\mathbb{P}_{(\mathbf{x},y) \in S_2}[|\langle \mathbf{w}, \mathbf{x} \rangle| \leq \theta] > C_1 \lambda^{C_1} \theta$ then **reject**.

   - Compute the $(d-1) \times (d-1)$ matrix $M_{S_2}$ as follows:[4]

$$ M_{S_2} = \mathop{\mathbb{E}}_{(\mathbf{x},y) \in S_2} \left[ \sum_{i=2}^{\infty} \frac{(\mathrm{proj}_{\perp \mathbf{w}} \mathbf{x})(\mathrm{proj}_{\perp \mathbf{w}} \mathbf{x})^T}{(i-1)^2} \mathbb{1}\{|\langle \mathbf{w}, \mathbf{x} \rangle| \in [(i-1)\theta, i\theta)\} \right] $$

   - If $\|M_S\|_{\mathrm{op}} > C_1 \theta \lambda^{C_1}$, then **reject**.

8. Otherwise, **accept** and output the vector $\mathbf{w}$ that achieves the smallest empirical error on $S_2$ among the vectors in the list $L$.

This concludes the algorithm. The full proof of Theorem 4.1 may be found in Appendix C.3.

---

[2] Or the same holds for the inverse vector.
[3] The operator $\mathrm{proj}_{\perp \mathbf{w}} : \mathbb{R}^d \to \mathbb{R}^{d-1}$ projects vectors on the hyperplane orthogonal to $\mathbf{w}$.
[4] Note that only at most $|S_2|$ many terms below are non-zero, hence $M_{S_2}$ can be computed efficiently.

## Acknowledgments and Disclosure of Funding

We wish to thank the anonymous reviewers of NeurIPS 2023 for their constructive feedback. Aravind Gollakota was at UT Austin while this work was done, supported by NSF award AF-1909204 and the NSF AI Institute for Foundations of Machine Learning (IFML). Adam R. Klivans was supported by NSF award AF-1909204 and the NSF AI Institute for Foundations of Machine Learning (IFML). Konstantinos Stavropoulos was supported by NSF award AF-1909204, the NSF AI Institute for Foundations of Machine Learning (IFML), and by scholarships from Bodossaki Foundation and Leventis Foundation. Arsen Vasilyan was supported in part by NSF awards CCF-2006664, DMS-2022448, CCF-1565235, CCF-1955217, CCF-2310818, Big George Fellowship and Fintech@CSAIL. Part of this work was done while Arsen Vasilyan was visiting UT Austin.

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

# A  Technical Lemmas

In this section, we provide a list of technical results that we use in our proofs.

**Lemma A.1** (Preservation of Poincaré constant). *Let $I$ be an open interval in $\mathbb{R}$ and $q : \mathbb{R}^d \to \mathbb{R}_+$ the density of a $\gamma$-Poincaré distribution. Let $\mathbf{v} \in \mathbb{S}^{d-1}$ and $q'_{\mathbf{v}} : \mathbb{R}^{d-1} \to \mathbb{R}_+$ be the density of the distribution resulting from conditioning $q$ to $\mathbf{x} \cdot \mathbf{v} \in I$ and projecting on the subspace perpendicular to $\mathbf{v}$. Then, the distribution corresponding to $q'_{\mathbf{v}}$ is $\gamma$-Poincaré.*

*Proof.* Assume, without loss of generality, that $\mathbf{v} = \mathbf{e}_d$. We have that

$$q'_{\mathbf{v}}(\mathbf{x}_{<d}) = \frac{\int_{x_d \in I} q(\mathbf{x}_{<d}, x_d) \, dx_d}{\int_{\mathbf{x}_{<d}} \int_{x_d \in I} q(\mathbf{x}) \, d\mathbf{x}} \ , \text{ for any } \mathbf{x}_{<d} \in \mathbb{R}^{d-1} \ .$$

Let $f : \mathbb{R}^{d-1} \to \mathbb{R}$ be any differentiable function. In order to show that $q'_{\mathbf{v}}$ is $\gamma$-Poincaré, it is sufficient to show that under no further assumptions on $f$, the quantity $\mathrm{Var}_{q'_{\mathbf{v}}}(f(\mathbf{x}_{<d}))$ is upper bounded by the product of $\gamma$ and $\mathbb{E}_{q'_{\mathbf{v}}}[\|\nabla f(\mathbf{x}_{<d})\|_2^2]$. We expand the quantity $\mathrm{Var}_{q'_{\mathbf{v}}}(f(\mathbf{x}_{<d}))$ as follows

$$
\begin{aligned}
\mathrm{Var}_{q'_{\mathbf{v}}}(f(\mathbf{x}_{<d})) &= \inf_\tau \int_{\mathbf{x}_{<d}} (f(\mathbf{x}_{<d}) - \tau)^2 q'_{\mathbf{v}}(\mathbf{x}_{<d}) \, d\mathbf{x}_{<d} \\
&= \inf_\tau \int_{\mathbf{x}_{<d}} (f(\mathbf{x}_{<d}) - \tau)^2 \cdot \frac{\int_{x_d \in I} q(\mathbf{x}_{<d}, x_d) \, dx_d}{\int_{\mathbf{x}_{<d}} \int_{x_d \in I} q(\mathbf{x}) \, d\mathbf{x}} \, d\mathbf{x}_{<d} \\
&= \frac{\inf_\tau \int_{\mathbf{x}_{<d}} \int_{x_d \in I} (f(\mathbf{x}_{<d}) - \tau)^2 \cdot q(\mathbf{x}) \, dx_d \, d\mathbf{x}_{<d}}{\int_{\mathbf{x}_{<d}} \int_{x_d \in I} q(\mathbf{x}) \, d\mathbf{x}} \\
&\leq \frac{\gamma \cdot \int_{\mathbf{x}_{<d}} \int_{x_d \in I} \|\nabla_{\mathbf{x}} f(\mathbf{x}_{<d})\|_2^2 \cdot q(\mathbf{x}) \, d\mathbf{x}}{\int_{\mathbf{x}_{<d}} \int_{x_d \in I} q(\mathbf{x}) \, d\mathbf{x}} \qquad \text{(since } q \text{ is } \gamma\text{-Poincaré)} \\
&= \gamma \cdot \int_{\mathbf{x}_{<d}} \|\nabla_{\mathbf{x}_{<d}} f(\mathbf{x}_{<d})\|_2^2 \cdot q'_{\mathbf{v}}(\mathbf{x}_{<d}) \, d\mathbf{x}_{<d} \qquad \text{(since } \frac{\partial f}{\partial x_d} \equiv 0\text{)} \\
&= \gamma \cdot \mathbb{E}_{q'_{\mathbf{v}}}[\|\nabla f(\mathbf{x}_{<d})\|_2^2] \ ,
\end{aligned}
$$

which concludes the proof. $\qquad\qquad\square$

**Proposition A.2** (Spectral Tester). *Let $D$ be a distribution over $\mathbb{R}^d$. Then, there is a tester that given $\delta \in (0,1)$, $\lambda \geq 1$, $\theta > 0$ and a set $S$ of i.i.d. samples from $D$ with size at least $\frac{2\lambda d^4}{\theta^2 \delta}$, runs in time $\mathrm{poly}(d, \frac{1}{\theta}, |S|)$ and satisfies the following specifications*

    *(a) If the tester accepts, then, for $\mathbf{z} \sim S$, $\mathbb{E}_S[\mathbf{z}\mathbf{z}^T] \succeq \frac{\theta}{2} I_d$ (resp. $\mathbb{E}_S[\mathbf{z}\mathbf{z}^T] \preceq 2\theta I_d$).*

    *(b) If, for $\mathbf{z} \sim D$, $\mathbb{E}_D[(\mathbf{z}_i \mathbf{z}_j)^2] \leq \lambda$ and $\mathbb{E}_D[\mathbf{z}\mathbf{z}^T] \succeq \theta I_d$ (resp. $\mathbb{E}_D[\mathbf{z}\mathbf{z}^T] \preceq \theta I_d$), then the tester accepts with probability at least $1 - \delta$.*

*Proof.* The tester receives $\lambda$, a set $S$ and $\delta \in (0,1)$ and does the following:

    1. Compute the matrix $M_S = \mathbb{E}_S[\mathbf{z}\mathbf{z}^T]$.

    2. If the minimum (resp. maximum) eigenvalue of $M_S$ is larger than $\frac{\theta}{2}$ (resp. smaller than $2\theta$), then **accept**. Otherwise **reject**.

Clearly, if the tester accepts, then the desired property is satisfied by construction. If the distribution $D$ satisfies the conditions of part (b), we can show that for $M_D = \mathbb{E}_{\mathbf{z} \sim D}[\mathbf{z}\mathbf{z}^T]$ we have

$$\left\| M_S - M_D \right\|_{\mathrm{op}} \leq \frac{\theta}{2}, \text{ with probability at least } 1 - \delta$$

which implies that $M_S \succeq \frac{\theta}{2} I_d$ (and $M_S \preceq (\theta + \frac{\theta}{2}) I_d \preceq 2\theta I_d$). In particular, we have that $(M_S)_{ij} = \mathbb{E}_S[\mathbf{z}_i \mathbf{z}_j]$, and by Chebyshev's inequality we have

$$\mathbb{P}\left[|(M_S)_{ij} - (M_D)_{ij}| > \frac{\theta}{2d}\right] \leq \frac{4d^2}{\theta^2 |S|} \mathbb{E}_{\mathbf{z} \sim D}[(\mathbf{z}_i \mathbf{z}_j)^2] \leq \frac{4\lambda d^2}{\theta^2 |S|} \leq \frac{\delta}{\binom{d}{2}}$$

By a union bound, we get that $\|M_S - M_D\|_{\max} \leq \frac{\theta}{2d}$ with probability at least $1 - \delta$ and hence $\|M_S - M_D\|_{\mathrm{op}} \leq d\|M_S - M_D\|_{\max} \leq \frac{\theta}{2}$, which concludes the proof. $\qquad\square$

**Proposition A.3.** *Let $c \geq 0$, $\lambda \geq 1$, $\sigma \leq \frac{1}{2\lambda}$ and $D$ be a $\lambda$-nice distribution over $\mathbb{R}^d$. Then, for any unit vectors $\mathbf{w}, \mathbf{v}, \mathbf{v}', \mathbf{u}, \mathbf{u}' \in \mathbb{R}^d$ with $\langle \mathbf{w}, \mathbf{v} \rangle = \langle \mathbf{w}, \mathbf{v}' \rangle = 0$ and for some universal constant $C > 0$ we have*

    *(i)* $\mathbb{P}[|\langle \mathbf{w}, \mathbf{x} \rangle| \leq \sigma] = 2\sigma \cdot \alpha^C$, *for some* $\alpha \in [\frac{1}{C\lambda}, C\lambda]$.

    *(ii)* $\mathbb{E}[\langle \mathbf{v}, \mathbf{x} \rangle^2 \cdot \mathbb{1}\{|\langle \mathbf{w}, \mathbf{x} \rangle| \leq \sigma\}] = 2\sigma \cdot \alpha^C$, *for some* $\alpha \in [\frac{1}{C\lambda}, C\lambda]$.

    *(iii)* $\mathbb{E}[\langle \mathbf{x}, \mathbf{u} \rangle^2 \langle \mathbf{x}, \mathbf{u}' \rangle^2] = \alpha^C$, *for some* $\alpha \leq C\lambda$.

    *(iv)* $\mathbb{E}[\langle \mathbf{v}, \mathbf{x} \rangle^2 \cdot \mathbb{1}\{|\langle \mathbf{w}, \mathbf{x} \rangle| \in [c, c + \sigma]\}] \leq 2\sigma \cdot \alpha^C$, *for some* $\alpha \leq C\lambda$.

*Proof.* We start by deriving property (i). Recall the function $Q$ from the definition of a $\lambda$-nice distribution, which upper-bounds the density of any two-dimensional projection of a $\lambda$-nice distribution we see that:

$$\mathbb{P}[|\langle \mathbf{w}, \mathbf{x} \rangle| \leq \sigma] = \int_{x_1 = -\sigma}^{\sigma} \int_{x_2 = -\infty}^{\infty} q_{\mathrm{span}(\mathbf{v}, \mathbf{w})}(x_1, x_2) \, dx_1 dx_2$$

$$\leq \int_{x_1 = -\sigma}^{\sigma} \int_{x_2 = -\infty}^{\infty} Q\left(\sqrt{x_1^2 + x_2^2}\right) dx_1 dx_2$$

Now, note that the region $\{(x_1, x_2) : |x_1| \leq \sigma\}$ is a subset of the set

$$\{(x_1, x_2) : |x_2| \leq \sigma |x_1|\} \cup \{(x_1, x_2) : |x_1| \leq \sigma \ \& \ |x_2| \leq 1\}.$$

Therefore:

$$\int_{x_1 = -\sigma}^{\sigma} \int_{x_2 = -\infty}^{\infty} Q\left(\sqrt{x_1^2 + x_2^2}\right) dx_1 dx_2 \leq$$

$$4 \arcsin(\sigma) \cdot \int_{r=0}^{\infty} 2\pi r Q(r) \, dr + \int_{x_1 = -\sigma}^{\sigma} \int_{x_2 = -1}^{1} Q\left(\sqrt{x_1^2 + x_2^2}\right) dx_1 dx_2 \leq O(\sigma\lambda)$$

Note that in the last line above, we bounded the first term via the bound $\int_{r=0}^{\infty} r Q(r) \, dr \leq \lambda$ from the definition of $\lambda$-nice distributions. Likewise, we bounded the second term via the inequality $Q(r) \leq \lambda$ from the definition of $\lambda$-nice distributions. Overall, we get

$$\mathbb{E}[\langle \mathbf{v}, \mathbf{x} \rangle^2 \cdot \mathbb{1}\{|\langle \mathbf{w}, \mathbf{x} \rangle| \leq \sigma\}] \leq O(\sigma\lambda)$$

Now, we shall lower-bound the same quantity. We have

$$\mathbb{P}[|\langle \mathbf{w}, \mathbf{x} \rangle| \leq \sigma] = \int_{x_1 = -\sigma}^{\sigma} \int_{x_2 = -\infty}^{\infty} q_{\mathrm{span}(\mathbf{v}, \mathbf{w})}(x_1, x_2) \, dx_1 dx_2$$

$$\geq \int_{x_1 = -\sigma}^{\sigma} \int_{x_2 = -\frac{1}{2\lambda}}^{\frac{1}{2\lambda}} q_{\mathrm{span}(\mathbf{v}, \mathbf{w})}(x_1, x_2) \, dx_1 dx_2$$

Now, since $\sigma \leq \frac{1}{2\lambda}$ via the premise of the lemma, we see that the whole region of integration on the right side of the set $\{(x_1, x_2) : \sqrt{x_1^2 + x_2^2} \leq \frac{1}{\lambda}\}$. From the definition of $\lambda$-nice distributions, the density $q_{\mathrm{span}(\mathbf{v}, \mathbf{w})}$ is lower-bounded by $1/\lambda$ in this region. Therefore, we have

$$\mathbb{P}[|\langle \mathbf{w}, \mathbf{x} \rangle| \leq \sigma] \geq \frac{2\sigma}{\lambda} \cdot \frac{1}{\lambda} = \frac{2\sigma}{\lambda^2},$$

which finishes the proof of property (i).

Now, we derive property (ii). Recall the function $Q$ from the definition of a $\lambda$-nice distribution, which upper-bounds the density of any two-dimensional projection of a $\lambda$-nice distribution we see that:

$$\mathbb{E}[\langle \mathbf{v}, \mathbf{x} \rangle^2 \cdot \mathbb{1}\{|\langle \mathbf{w}, \mathbf{x} \rangle| \leq \sigma\}] = \int_{x_1=-\sigma}^{\sigma} \int_{x_2=-\infty}^{\infty} x_2^2 \cdot q_{\text{span}(\mathbf{v}, \mathbf{w})}(x_1, x_2) \, dx_1 dx_2$$

$$\leq \int_{x_1=-\sigma}^{\sigma} \int_{x_2=-\infty}^{\infty} x_2^2 \cdot Q\left(\sqrt{x_1^2 + x_2^2}\right) dx_1 dx_2$$

Now, note that the region $\{(x_1, x_2) : |x_1| \leq \sigma\}$ is a subset of the set

$$\{(x_1, x_2) : |x_2| \leq \sigma|x_1|\} \cup \{(x_1, x_2) : |x_1| \leq \sigma \ \& \ |x_2| \leq 1\}.$$

Therefore:

$$\int_{x_1=-\sigma}^{\sigma} \int_{x_2=-\infty}^{\infty} x_2^2 \cdot Q\left(\sqrt{x_1^2 + x_2^2}\right) dx_1 dx_2 \leq$$

$$4 \arcsin(\sigma) \cdot \int_{r=0}^{\infty} 2\pi r^3 Q(r) \, dr + \int_{x_1=-\sigma}^{\sigma} \int_{x_2=-1}^{1} x_2^2 \cdot Q\left(\sqrt{x_1^2 + x_2^2}\right) dx_1 dx_2 \leq O(\sigma\lambda)$$

Note that in the last line above, we bounded the first term via the bound on $\int_{r=0}^{\infty} r^3 Q(r) \, dr$ from the definition of $\lambda$-nice distributions. Likewise, we bounded the second term via the inequality $Q(r) \leq \lambda$ from the definition of $\lambda$-nice distributions. Therefore, we get

$$\mathbb{E}[\langle \mathbf{v}, \mathbf{x} \rangle^2 \cdot \mathbb{1}\{|\langle \mathbf{w}, \mathbf{x} \rangle| \leq \sigma\}] \leq O(\sigma\lambda)$$

Now, we shall lower-bound the same quantity. We have

$$\mathbb{E}[\langle \mathbf{v}, \mathbf{x} \rangle^2 \cdot \mathbb{1}\{|\langle \mathbf{w}, \mathbf{x} \rangle| \leq \sigma\}] = \int_{x_1=-\sigma}^{\sigma} \int_{x_2=-\infty}^{\infty} x_2^2 \cdot q_{\text{span}(\mathbf{v}, \mathbf{w})}(x_1, x_2) \, dx_1 dx_2$$

$$\geq \int_{x_1=-\sigma}^{\sigma} \int_{x_2=-\frac{1}{2\lambda}}^{\frac{1}{2\lambda}} x_2^2 \cdot q_{\text{span}(\mathbf{v}, \mathbf{w})}(x_1, x_2) \, dx_1 dx_2$$

Now, since $\sigma \leq \frac{1}{2\lambda}$ via the premise of the lemma, we see that the whole region of integration on the right side of the set $\{(x_1, x_2) : \sqrt{x_1^2 + x_2^2} \leq \frac{1}{\lambda}\}$. From the definition of $\lambda$-nice distributions, the density $q_{\text{span}(\mathbf{v}, \mathbf{w})}$ is lower-bounded by $1/\lambda$ in this region. Therefore, we have

$$\mathbb{E}[\langle \mathbf{v}, \mathbf{x} \rangle^2 \cdot \mathbb{1}\{|\langle \mathbf{w}, \mathbf{x} \rangle| \leq \sigma\}] \geq \frac{2\sigma}{\lambda} \cdot \frac{1}{4\lambda^2} \cdot \frac{1}{\lambda} = \frac{\sigma}{2\lambda^4},$$

which finishes the proof of property (ii).

We proceed to property (iii). We will denote the angle between $\mathbf{v}$ and $\mathbf{v}'$ as $\beta$, which allows us to write

$$\mathbb{E}[\langle \mathbf{x}, \mathbf{v} \rangle^2 \langle \mathbf{x}, \mathbf{v}' \rangle^2] = \int_{x_1=-\infty}^{\infty} \int_{x_2=-\infty}^{\infty} x_1^2(x_1 \cos\beta + x_2 \sin\beta)^2 q_{\text{span}(\mathbf{v}, \mathbf{w})}(x_1, x_2) \, dx_1 dx_2$$

$$\leq \int_{x_1=-\infty}^{\infty} \int_{x_2=-\infty}^{\infty} x_1^2(x_1 \cos\beta + x_2 \sin\beta)^2 \cdot Q\left(\sqrt{x_1^2 + x_2^2}\right) dx_1 dx_2$$

$$\leq \int_{x_1=-\infty}^{\infty} \int_{x_2=-\infty}^{\infty} (x_1^2 + x_2^2)^2 \cdot Q\left(\sqrt{x_1^2 + x_2^2}\right) dx_1 dx_2$$

$$= \int_{r=0}^{\infty} 2\pi r^5 Q(r) \, dr \leq 2\pi\lambda,$$

which finishes the proof of property (iii).

Finally, we prove property (iv). For $\beta \geq 0$ we have

$$\int_{r=0}^{\infty} r^2 Q\big(\sqrt{r^2+\beta}\big)\, dr = \int_{r=0}^{1} r^2 Q\big(\sqrt{r^2+\beta}\big)\, dr + \int_{r=1}^{\infty} r^2 Q\big(\sqrt{r^2+\beta}\big)\, dr$$

$$\leq \lambda + \int_{r=1}^{\infty} r^3 Q\big(\sqrt{r^2+\beta}\big)\, dr \qquad (\text{since } \sup_{r\geq 0} Q(r) \leq \lambda)$$

$$\leq \lambda + \int_{r'=\sqrt{1+\beta}}^{\infty} (r'^3 - \beta r') Q(r')\, dr' \qquad (\text{by setting } r' = \sqrt{r^2+\beta})$$

$$\leq \lambda + \int_{r=0}^{\infty} r^3 Q(r)\, dr \qquad (\text{since } \beta r Q(r) \geq 0 \text{ for any } r \geq 0)$$

$$\leq 2\lambda$$

Applying the above inequality to the quantity of property (iv), we get the desired result.

$$\mathbb{E}[\langle \mathbf{v}, \mathbf{x}\rangle^2 \cdot \mathbb{1}\{|\langle \mathbf{w}, \mathbf{x}\rangle| \in [c, c+\sigma]\}] = \int_{|x_1|\in[c,c+\sigma]} \int_{x_2=-\infty}^{\infty} x_2^2 \cdot q_{\mathrm{span}(\mathbf{v},\mathbf{w})}\, dx_1 dx_2$$

$$\leq \int_{|x_1|\in[c,c+\sigma]} \int_{x_2=-\infty}^{\infty} x_2^2 \cdot Q\big(\sqrt{x_1^2+x_2^2}\big)\, dx_1 dx_2$$

$$= \int_{|x_1|\in[c,c+\sigma]} \left( 2\int_{r=0}^{\infty} r^2 \cdot Q\big(\sqrt{x_1^2+r^2}\big)\, dr \right) dx_1$$

$$\leq \int_{|x_1|\in[c,c+\sigma]} (4\lambda)\, dx_1 \leq 8\lambda\sigma$$

This concludes the proof of Proposition A.3. $\qquad\square$

**Proposition A.4** (Proposition 4.2 of [GKSV23]). *There is a universal constant $C > 0$, such that for any $\sigma > 0$, there exists a continuously differentiable function $\ell_\sigma : \mathbb{R} \to [0,1]$ with the following properties.*

1. *For any $t \in [-\sigma/6, \sigma/6]$, $\ell_\sigma(t) = \frac{1}{2} + \frac{t}{\sigma}$.*

2. *For any $t > \sigma/2$, $\ell_\sigma(t) = 1$ and for any $t < -\sigma/2$, $\ell_\sigma(t) = 0$.*

3. *For any $t \in \mathbb{R}$, $\ell'_\sigma(t) \in [0, C/\sigma]$, $\ell'_\sigma(t) = \ell'_\sigma(-t)$ and $|\ell''_\sigma(t)| \leq C/\sigma^2$.*

**Proposition A.5** (PSGD Convergence [DKTZ20a], restated in [GKSV23]). *Let $\mathcal{L}_\sigma$ be as in Equation (4.1) with $\sigma \in (0,1]$, $\ell_\sigma$ as described in Proposition A.4, $\lambda \geq 1$ and $D_{\mathcal{XY}}$ such that the marginal $D_{\mathcal{X}}$ on $\mathbb{R}^d$ is $\lambda$-nice. Then for some universal constant $C > 0$ and for any $\epsilon > 0$ and $\delta \in (0,1)$, there is an algorithm whose time and sample complexity is $O(\frac{\lambda^C d}{\sigma^4} + \frac{\lambda^C \log(1/\delta)}{\epsilon^4 \sigma^4})$, which, having access to samples from $D_{\mathcal{XY}}$, outputs a list $L$ of vectors $\mathbf{w} \in \mathbb{S}^{d-1}$ with $|L| = O(\frac{\lambda^C d}{\sigma^4} + \frac{\lambda^C \log(1/\delta)}{\epsilon^4 \sigma^4})$ so that there exists $\mathbf{w} \in L$ with*

$$\|\nabla_{\mathbf{w}} \mathcal{L}_\sigma(\mathbf{w}; D_{\mathcal{XY}})\|_2 \leq \epsilon, \text{ with probability at least } 1 - \delta.$$

*In particular, the algorithm performs Stochastic Gradient Descent on $\mathcal{L}_\sigma$ Projected on $\mathbb{S}^{d-1}$ (PSGD).*

# B    Proofs from Section 3

## B.1    Proof of Lemma 3.1

We restate Lemma 3.1 here for convenience.

**Lemma B.1** (Lemma 3.1). *Let $D_{\mathcal{XY}}$ be a distribution over $\mathbb{R}^d \times \{\pm 1\}$, $\mathbf{w} \in \mathbb{S}^{d-1}$, $\theta \in (0, \pi/4]$, $\lambda \geq 1$ and $\delta \in (0,1)$. Then, for a sufficiently large constant $C$, there is a tester that given $\delta$, $\theta$, $\mathbf{w}$ and a set $S$ of samples from $D_{\mathcal{X}}$ with size at least $C \cdot \left( \frac{d^4}{\theta^2 \delta} \right)$, runs in time $\mathrm{poly}\left(d, \frac{1}{\theta}, \frac{1}{\delta}\right)$ and satisfies the following specifications:*

(a) *If the tester accepts $S$, then for every unit vector $\mathbf{w}' \in \mathbb{R}^n$ satisfying $\angle(\mathbf{w}, \mathbf{w}') \leq \theta$ we have*

$$\mathbb{P}_{\mathbf{x} \sim S}[\mathrm{sign}(\langle \mathbf{w}', \mathbf{x} \rangle) \neq \mathrm{sign}(\langle \mathbf{w}, \mathbf{x} \rangle)] \leq C \cdot \theta \cdot \lambda^C$$

(b) *If the distribution $D_{\mathcal{X}}$ is $\lambda$-nice, the tester accepts $S$ with probability $1 - \delta$.*

*Proof.* The testing algorithm receives integer $d$, set $S \subset \mathbb{R}^d$, $\mathbf{w} \in \mathbb{S}^{d-1}$, $\theta \in (0, \pi/4]$, $\lambda \geq 1$ and $\delta \in (0, 1)$ and does the following for some sufficiently large universal constant $C_1 > 0$:

1. If $\mathbb{P}_{\mathbf{x} \in S}[|\langle \mathbf{w}, \mathbf{x} \rangle| \in [0, \theta]] > C_1 \theta \lambda^{C_1}$, then **reject**.

2. Let $\mathrm{proj}_{\perp \mathbf{w}} : \mathbb{R}^d \to \mathbb{R}^{d-1}$ denote the operator that given any vector in $\mathbb{R}^d$, it outputs its projection into the $(d-1)$-dimensional subspace of $\mathbb{R}^d$ that is orthogonal to $\mathbf{w}$.

3. Compute the $(d-1) \times (d-1)$ matrix $M_S$ as follows[5]:

$$M_S = \mathbb{E}_{\mathbf{x} \in S}\left[ \sum_{i=2}^{\infty} \frac{(\mathrm{proj}_{\perp \mathbf{w}} \mathbf{x})(\mathrm{proj}_{\perp \mathbf{w}} \mathbf{x})^T}{(i-1)^2} \mathbb{1}\{|\langle \mathbf{w}, \mathbf{x} \rangle| \in [(i-1)\theta, i\theta)\} \right]$$

4. Run the spectral tester of Proposition A.2 on $M_S$ given $\delta \leftarrow \delta$, $\lambda \leftarrow C_1 \lambda^{C_1}$ and $\theta \leftarrow \frac{C_1}{2} \theta \lambda^{C_1}$, i.e., compute $\|M_S\|_{\mathrm{op}}$ and if $\|M_S\|_{\mathrm{op}} > C_1 \theta \lambda^{C_1}$, then **reject**. Otherwise, **accept**.

First, suppose the test accepts. For the following, consider the vector $\mathbf{w}' \in \mathbb{R}^d$ to be an arbitrary unit vector and $\mathbf{v} \in \mathbb{R}^d$ to be the unit vector that is perpendicular to $\mathbf{w}$, lies within the plane defined by $\mathbf{w}$ and $\mathbf{w}'$ and $\langle \mathbf{v}, \mathbf{w}' \rangle \leq 0$. Then we have:

$$\mathbb{P}_{\mathbf{x} \sim S}[\mathrm{sign}(\langle \mathbf{w}', \mathbf{x} \rangle) \neq \mathrm{sign}(\langle \mathbf{w}, \mathbf{x} \rangle)] \leq$$

$$\leq \sum_{i=1}^{\infty} \mathbb{P}_{\mathbf{x} \sim S}\Big[ \underbrace{|\langle \mathbf{v}, \mathbf{x} \rangle| > \frac{\theta}{\tan \theta} \cdot (i-1)}_{\text{Implies } |\langle \mathbf{v}, \mathbf{x} \rangle| > (i-1)/2} \ \& \ |\langle \mathbf{w}, \mathbf{x} \rangle| \in [(i-1)\theta, i\theta] \Big]$$

$$\leq \underbrace{\mathbb{P}_{\mathbf{x} \in S}[|\langle \mathbf{w}, \mathbf{x} \rangle| \in [0, \theta]]}_{\leq C_1 \theta \lambda^{C_1}} + 4 \underbrace{\sum_{i=2}^{\infty} \frac{\mathbb{E}_{\mathbf{x} \sim S}\left[ \langle \mathbf{v}, \mathbf{x} \rangle^2 \mathbb{1}_{|\langle \mathbf{w}, \mathbf{x} \rangle| \in [(i-1)\theta, i\theta]} \right]}{(i-1)^2}}_{\langle \mathrm{proj}_{\perp \mathbf{w}} \mathbf{v}, M \, \mathrm{proj}_{\perp \mathbf{w}} \mathbf{v} \rangle \leq \|M\|_{\mathrm{op}} \leq C_1 \theta \lambda^{C_1}}$$

$$\leq 5 C_1 \theta \lambda^{C_1}$$

For part (b), we suppose that the distribution $D_{\mathcal{X}}$ is indeed $\lambda$-nice. We will show that with probability at least $1 - \delta$, the tester will accept, i.e., that

$$\mathbb{P}_{\mathbf{x} \in S}[|\langle \mathbf{w}, \mathbf{x} \rangle| \in [0, \theta]] \leq C_1 \theta \lambda^{C_1} \text{ and} \tag{B.1}$$

$$\|M_S\|_{\mathrm{op}} \leq C_1 \theta \lambda^{C_1} \tag{B.2}$$

We first observe that the corresponding quantities under distribution $D_{\mathcal{X}}$ due to Proposition A.3. In particular, we have that for some universal constant $C' > 0$

$$\mathbb{P}_{\mathbf{x} \in D_{\mathcal{X}}}[|\langle \mathbf{w}, \mathbf{x} \rangle| \in [0, \theta]] \leq C' \theta \lambda^{C'} \text{ and} \tag{B.3}$$

$$\mathbb{E}_{\mathbf{x} \in D_{\mathcal{X}}}[\langle \mathbf{v}', \mathbf{x} \rangle^2 \cdot \mathbb{1}\{|\langle \mathbf{w}, \mathbf{x} \rangle| \in [c, c+\theta]\}] \leq C' \theta \lambda^{C'} \text{ for any } \mathbf{v}' \in \mathbb{S}^{d-1} \text{ and } c \geq 0 \tag{B.4}$$

If we let $M_{D_{\mathcal{X}}} = \mathbb{E}_{D_{\mathcal{X}}}[M_S]$, we get that

$$\|M_{D_{\mathcal{X}}}\|_{\mathrm{op}} = \sup_{\mathbf{u} \in \mathbb{S}^{d-2}} \mathbf{u}^T M_{D_{\mathcal{X}}} \mathbf{u} = \sup_{\mathbf{v}' \in \mathbb{S}^{d-1} : \langle \mathbf{v}', \mathbf{w} \rangle = 0} (\mathrm{proj}_{\perp \mathbf{w}} \mathbf{v}')^T M_{D_{\mathcal{X}}} (\mathrm{proj}_{\perp \mathbf{w}} \mathbf{v}')$$

$$\leq \sum_{i=2}^{\infty} \frac{1}{(i-1)^2} \sup_{\mathbf{v}' \in \mathbb{S}^{d-1}} \mathbb{E}_{\mathbf{x} \in D_{\mathcal{X}}}[\langle \mathbf{v}', \mathbf{x} \rangle^2 \cdot \mathbb{1}\{|\langle \mathbf{w}, \mathbf{x} \rangle| \in [c, c+\theta]\}]$$

$$\leq \sum_{i=2}^{\infty} \frac{1}{(i-1)^2} \sup_{\mathbf{v}' \in \mathbb{S}^{d-1}} C' \theta \lambda^{C'} \leq \frac{C' \pi^2}{6} \theta \lambda^{C'}$$

---

[5]Note that only at most $|S|$ many terms below are non-zero, hence $M_S$ can be computed efficiently.

By Proposition A.2, in order to satisfy expression (B.2), it remains to show that $\mathbb{E}_{\mathbf{z} \sim D}[(\mathbf{z}_\ell \mathbf{z}_j)^2] \leq C_1 \lambda^{C_1}$ for any $\ell, j \in [d]$, where $\mathbf{z}$ is defined as follows

$$\mathbf{z} = \sum_{i=2}^{\infty} \frac{\mathrm{proj}_{\perp \mathbf{w}} \mathbf{x}}{(i-1)} \mathbb{1}_{|\langle \mathbf{w}, \mathbf{x} \rangle| \in [(i-1)\theta, i\theta)}.$$

Since $\mathbb{E}_{\mathbf{z} \sim D}[(\mathbf{z}_\ell \mathbf{z}_j)^2] \leq \mathbb{E}_{\mathbf{z} \sim D}[\langle \mathbf{u}, \mathbf{x} \rangle^2 \langle \mathbf{u}', \mathbf{x} \rangle^2]$, for some unit vectors $\mathbf{u}, \mathbf{u}' \in \mathbb{S}^{d-1}$ (orthogonal to $\mathbf{w}$), the desired bound follows from Proposition A.3.

It remains to bound the absolute distance between the quantities of the left hand side of expressions (B.1) and (B.3). This can be achieved by an application of the Hoeffding bound, since the empirical version of the quantity is the average of independent Bernoulli random variables. $\qquad \square$

## B.2 Proof of Lemma 3.2

We restate Lemma 3.2 here for convenience.

**Lemma B.2** (Universally Testable Weak Anti-Concentration). *Let $D$ be a distribution over $\mathbb{R}^d$. Then, there is a universal constant $C > 0$ and a tester that given a unit vector $\mathbf{w} \in \mathbb{R}^d$, $\delta \in (0, 1)$, $\gamma > 0$, $\lambda \geq 1$, $\sigma \leq \frac{1}{2\lambda}$ and a set $S$ of i.i.d. samples from $D$ with size at least $C \cdot \frac{d^4}{\sigma^2 \delta} \log(d) \lambda^C$, runs in time $\mathrm{poly}(d, \lambda, \frac{1}{\sigma}, \frac{1}{\delta}, \log(\frac{1}{\gamma}))$ and satisfies the following specifications*

*(a) If the tester accepts $S$, then for any unit vector $\mathbf{v} \in \mathbb{R}^d$ with $\langle \mathbf{v}, \mathbf{w} \rangle = 0$ we have*

$$\mathbb{P}_{\mathbf{x} \in S}\left[|\langle \mathbf{v}, \mathbf{x} \rangle| \geq \frac{1}{C\lambda^C} \,\middle|\, |\langle \mathbf{w}, \mathbf{x} \rangle| \leq \sigma\right] \geq \frac{1}{C\lambda^C \gamma^4}$$

*(b) If $D$ is $\gamma$-Poincaré and $\lambda$-nice, then the tester accepts $S$ with probability at least $1 - \delta$.*

*Proof.* The testing algorithm receives a set $S \subset \mathbb{R}^d$, $\mathbf{w} \in \mathbb{S}^{d-1}$, $\delta \in (0, 1)$, $\gamma > 0$, $\lambda \geq 1$ and $\sigma \leq \frac{1}{2\lambda}$ and does the following for some sufficiently large $C_1 > 0$:

1. If $\mathbb{P}_{\mathbf{x} \in S}[|\langle \mathbf{w}, \mathbf{x} \rangle| \leq \sigma] > 2\sigma \cdot C_1 \lambda^{C_1}$, then **reject**.

2. Compute the $(d-1) \times (d-1)$ matrix $M_S$ as follows:
$$M_S = \mathbb{E}_{\mathbf{x} \in S}\left[(\mathrm{proj}_{\perp \mathbf{w}} \mathbf{x})(\mathrm{proj}_{\perp \mathbf{w}} \mathbf{x})^T \cdot \mathbb{1}\{|\langle \mathbf{w}, \mathbf{x} \rangle \leq \sigma|\}\right]$$

3. Run the spectral tester of Proposition A.2 on $M_S$ given $\delta \leftarrow \delta$, $\lambda \leftarrow C_1 \lambda^{C_1}$ and $\theta \leftarrow \frac{2\sigma}{C_1 \lambda^{C_1}}$, i.e., **reject** if the minimum singular value of $M_S$ is less than $\frac{2\sigma}{C_1 \lambda^{C_1}}$.

4. Run the hypercontractivity tester (Prop. 3.5) on $S' = \{\mathrm{proj}_{\perp \mathbf{w}} \mathbf{x} : \mathbf{x} \in S$ and $|\langle \mathbf{w}, \mathbf{x} \rangle| \leq \sigma\}$, i.e., solve an appropriate SDP and **reject** if the solution is larger than a specified threshold. Otherwise, **accept**.

For part (a), we apply the Paley–Zygmund inequality to the random variable $Z = \langle \mathbf{v}, \mathbf{x} \rangle^2$ condtitioned on $|\langle \mathbf{w}, \mathbf{x} \rangle| \leq \sigma$ and get

$$\mathbb{P}_{\mathbf{x} \in S}\left[\langle \mathbf{v}, \mathbf{x} \rangle^2 \geq \frac{1}{2} \mathbb{E}_{\mathbf{x} \in S}\left[\langle \mathbf{v}, \mathbf{x} \rangle^2 \,\middle|\, |\langle \mathbf{w}, \mathbf{x} \rangle| \leq \sigma\right] \,\middle|\, |\langle \mathbf{w}, \mathbf{x} \rangle| \leq \sigma\right] \geq \frac{(\mathbb{E}_{\mathbf{x} \in S}[\langle \mathbf{v}, \mathbf{x} \rangle^2 \mid |\langle \mathbf{w}, \mathbf{x} \rangle| \leq \sigma])^2}{4 \mathbb{E}_{\mathbf{x} \in S}[\langle \mathbf{v}, \mathbf{x} \rangle^4 \mid |\langle \mathbf{w}, \mathbf{x} \rangle| \leq \sigma]}$$

Note that since $\langle \mathbf{v}, \mathbf{w} \rangle = 0$, we have $\langle \mathbf{v}, \mathbf{x} \rangle = \langle \mathrm{proj}_{\perp \mathbf{w}} \mathbf{v}, \mathrm{proj}_{\perp \mathbf{w}} \mathbf{x} \rangle$ (where $\|\mathbf{v}\|_2 = \|\mathrm{proj}_{\perp \mathbf{w}} \mathbf{v}\|_2$). Therefore, since $S$ has passed the spectral tester as well as the tester for the probability of lying within the strip $|\langle \mathbf{w}, \mathbf{x} \rangle| \leq \sigma$, we have that

$$\mathbb{E}_{\mathbf{x} \in S}\left[\langle \mathbf{v}, \mathbf{x} \rangle^2 \,\middle|\, |\langle \mathbf{w}, \mathbf{x} \rangle| \leq \sigma\right] = \frac{\mathbb{E}_{\mathbf{x} \in S}\left[\langle \mathbf{v}, \mathbf{x} \rangle^2 \cdot \mathbb{1}\{|\langle \mathbf{w}, \mathbf{x} \rangle| \leq \sigma\}\right]}{\mathbb{P}_{\mathbf{x} \in S}[|\langle \mathbf{w}, \mathbf{x} \rangle| \leq \sigma]} \geq \frac{1}{2C_1 \lambda^{2C_1}}$$

Moreover, $\{\mathbf{x} \in S : |\langle \mathbf{w}, \mathbf{x} \rangle| \leq \sigma\}$ has passed the hypercontractivity tester, and therefore, according to Proposition 3.5 we have

$$\mathbb{E}_{\mathbf{x} \in S}\left[\langle \mathbf{v}, \mathbf{x} \rangle^4 \,\middle|\, |\langle \mathbf{w}, \mathbf{x} \rangle| \leq \sigma\right] \leq C_1 \cdot \gamma^4$$

Combining the above inequalities we conclude the proof of part (a).

For part (b), we assume that $D$ is indeed $\lambda$-nice and $\gamma$-Poincaré. We first use Proposition A.3 as well as a Hoeffding bound, to get that $\mathbb{P}_{\mathbf{x}\in S}[|\langle\mathbf{w},\mathbf{x}\rangle|\leq\sigma]\in[\frac{2\sigma}{C'\lambda^{C'}},2\sigma\cdot C'\lambda^{C'}]$ with probability at least $1-\delta/3$ over $S$ (since $|S|$ is large enough), for some universal constant $C'>0$. Then, we use part (ii) of Proposition A.3 to lower bound the minimum eigenvalue of $M_D=\mathbb{E}_D[M_S]$ by $\frac{4\sigma}{C'\lambda^{C'}}$. Using part (iii) of Proposition A.3 to bound the second moment of each of the elements of $M_D$, we may use Proposition A.2 to get that $M_S\succeq\frac{2\sigma}{C'\lambda^{C'}}I_{d-1}$ (and our spectral test passes) with probability at least $1-\delta/3$. It remains to show that the hypercontractivity tester will accept with probability at least $1-\delta/3$ (since, then, the result follows from a union bound).

We acquire samples from the hypercontractivity tester through rejection sampling (we keep only the samples within the strip). Since the probability of falling inside the strip is at least $\frac{2\sigma}{C'\lambda^{C'}}$, the number of samples we will keep is at least $|S'|\geq\frac{|S|\sigma}{C''\lambda^{C'}}$, for some large enough constant $C''>0$ (due to Chernoff bound) and with probability at least $1-\delta/6$. We now apply Lemma A.1 to get that the distribution of $\mathrm{proj}_{\perp\mathbf{w}}\,\mathbf{x}$ conditioned on the strip $|\langle\mathbf{w},\mathbf{x}\rangle|\leq\sigma$ is $\gamma$-Poincaré, since $D$ is also $\gamma$-Poincaré. Hence, the hypercontractivity tester accepts with probability at least $1-\delta/6$ due to Proposition 3.5. $\qquad\square$

## C  Proofs from Section 4

### C.1  Proof of Proposition 4.2

We restate Proposition 4.2 here for completeness.

**Proposition C.1** (Modification from [GKSV23, DKTZ20a, DKTZ20b]). *For a distribution $D_{\mathcal{X}\mathcal{Y}}$ over $\mathbb{R}^d\times\{\pm1\}$ let* opt *be the minimum error achieved by some origin-centered halfspace and $\mathbf{w}^*\in\mathbb{S}^{d-1}$ a corresponding vector. Consider $\mathcal{L}_\sigma$ as in Equation (4.1) for $\sigma>0$ and let $\eta<1/2$. Let $\mathbf{w}\in\mathbb{S}^{d-1}$ with $\measuredangle(\mathbf{w},\mathbf{w}^*)=\theta<\frac{\pi}{2}$ and $\mathbf{v}\in\mathrm{span}(\mathbf{w},\mathbf{w}^*)$ such that $\langle\mathbf{v},\mathbf{w}\rangle=0$ and $\langle\mathbf{v},\mathbf{w}^*\rangle<0$. Then, for some universal constant $C>0$ and any $\alpha\geq\frac{\sigma}{2\tan\theta}$ we have $\|\nabla_{\mathbf{w}}\mathcal{L}_\sigma(\mathbf{w};D_{\mathcal{X}\mathcal{Y}})\|_2\geq A_1-A_2-A_3$, where*

$$A_1=\frac{\alpha}{C\cdot\sigma}\cdot\mathbb{P}\left[|\langle\mathbf{v},\mathbf{x}\rangle|\geq\alpha\ \ and\ \ |\langle\mathbf{w},\mathbf{x}\rangle|\leq\frac{\sigma}{6}\right]$$

$$A_2=\frac{C}{\tan\theta}\cdot\mathbb{P}\left[|\langle\mathbf{w},\mathbf{x}\rangle|\leq\frac{\sigma}{2}\right]\ \ and\ \ A_3=\frac{C}{\sigma}\cdot\sqrt{\mathrm{opt}}\cdot\sqrt{\mathbb{E}\left[\langle\mathbf{v},\mathbf{x}\rangle^2\cdot\mathbb{1}_{\{|\langle\mathbf{w},\mathbf{x}\rangle|\leq\frac{\sigma}{2}\}}\right]}$$

*Moreover, if the noise is Massart with rate $\eta$, then $\|\nabla_{\mathbf{w}}\mathcal{L}_\sigma(\mathbf{w};D_{\mathcal{X}\mathcal{Y}})\|_2\geq(1-2\eta)A_1-A_2$.*

*Proof.* The proof is a slight modification of a part of the proof of Lemma 4.4 in [GKSV23], but we present it here for completeness.

For any vector $\mathbf{x}\in\mathbb{R}^d$, let: $\mathbf{x}_{\mathbf{w}}=\langle\mathbf{w},\mathbf{x}\rangle$ and $\mathbf{x}_{\mathbf{v}}=\langle\mathbf{v},\mathbf{x}\rangle$. It follows that $\mathrm{proj}_V(\mathbf{x})=\mathbf{x}_{\mathbf{v}}\mathbf{e}_1+\mathbf{x}_{\mathbf{w}}\mathbf{e}_2$, where $\mathrm{proj}_V$ is the operator that orthogonally projects vectors on $V$. Using the fact that $\nabla_{\mathbf{w}}(\langle\mathbf{w},\mathbf{x}\rangle/\|\mathbf{w}\|_2)=\mathbf{x}-\langle\mathbf{w},\mathbf{x}\rangle\mathbf{w}=\mathbf{x}-\mathbf{x}_{\mathbf{w}}\mathbf{w}$ for any $\mathbf{w}\in\mathbb{S}^{d-1}$, the interchangeability of the gradient and expectation operators and the fact that $\ell'_\sigma$ is an even function we get that

$$\nabla_{\mathbf{w}}\mathcal{L}_\sigma(\mathbf{w})=\mathbb{E}\Big[-\ell'_\sigma(|\langle\mathbf{w},\mathbf{x}\rangle|)\cdot y\cdot(\mathbf{x}-\mathbf{x}_{\mathbf{w}}\mathbf{w})\Big]$$

Since the projection operator $\mathrm{proj}_V$ is a contraction, we have $\|\nabla_{\mathbf{w}}\mathcal{L}_\sigma(\mathbf{w})\|_2\geq\|\mathrm{proj}_V\nabla_{\mathbf{w}}\mathcal{L}_\sigma(\mathbf{w})\|_2$, and we can therefore restrict our attention to a simpler, two dimensional problem. In particular, since $\mathrm{proj}_V(\mathbf{x})=\mathbf{x}_{\mathbf{v}}\mathbf{e}_1+\mathbf{x}_{\mathbf{w}}\mathbf{e}_2$, we get

$$\|\mathrm{proj}_V\nabla_{\mathbf{w}}\mathcal{L}_\sigma(\mathbf{w})\|_2=\Big|\mathbb{E}\Big[-\ell'_\sigma(|\mathbf{x}_{\mathbf{w}}|)\cdot y\cdot\mathbf{x}_{\mathbf{v}}\Big]\Big|$$

$$=\Big|\mathbb{E}\Big[-\ell'_\sigma(|\mathbf{x}_{\mathbf{w}}|)\cdot\mathrm{sign}(\langle\mathbf{w}^*,\mathbf{x}\rangle)\cdot(1-2\,\mathbb{1}\{y\neq\mathrm{sign}(\langle\mathbf{w}^*,\mathbf{x}\rangle)\})\cdot\mathbf{x}_{\mathbf{v}}\Big]\Big|$$

Let $F(y,\mathbf{x})$ denote $1-2\,\mathbb{1}\{y\neq\mathrm{sign}(\langle\mathbf{w}^*,\mathbf{x}\rangle)\}$. We may write $\mathbf{x}_{\mathbf{v}}$ as $|\mathbf{x}_{\mathbf{v}}|\cdot\mathrm{sign}(\mathbf{x}_{\mathbf{v}})$ and let $\mathcal{G}\subseteq\mathbb{R}^2$ such that $\mathrm{sign}(\mathbf{x}_{\mathbf{v}})\cdot\mathrm{sign}(\langle\mathbf{w}^*,\mathbf{x}\rangle)=-1$ iff $\mathbf{x}\in\mathcal{G}$.

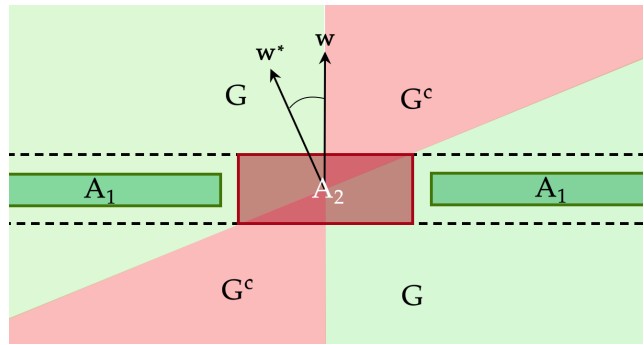

Figure 1: The Gaussian mass in each of the regions labelled $A_1$ and $A_2$ is proportional to the corresponding term appearing in the statement of Proposition 4.2. As $\sigma$ tends to $0$, the Gaussian mass of region $A_2$ shrinks faster than the one of region $A_1$, since both the height ($\sigma$) and the width $\left(\frac{\sigma}{\tan\theta}\right)$ of $A_2$ are proportional to $\sigma$, while the width of $A_1$ is not affected (the height is $\sigma/3$). Lemma 4.3 demonstrates that a similar property is universally testable under any nice Poincaré distribution.

Then, $\text{sign}(\mathbf{x_v}) \cdot \text{sign}(\langle \mathbf{w}^*, \mathbf{x} \rangle) = \mathbb{1}\{\mathbf{x} \notin \mathcal{G}\} - \mathbb{1}\{\mathbf{x} \in \mathcal{G}\}$. We get

$$
\begin{aligned}
\| \text{proj}_V \nabla_\mathbf{w} \mathcal{L}_\sigma(\mathbf{w}) \|_2 &= \\
&= \left| \mathbb{E}\Big[ \ell_\sigma'(|\mathbf{x_w}|) \cdot (\mathbb{1}\{\mathbf{x} \in \mathcal{G}\} - \mathbb{1}\{\mathbf{x} \notin \mathcal{G}\}) \cdot F(y, \mathbf{x}) \cdot |\mathbf{x_v}| \cdot \Big] \right| \\
&\geq \mathbb{E}\Big[ \ell_\sigma'(|\mathbf{x_w}|) \cdot \mathbb{1}\{\mathbf{x} \in \mathcal{G}\} \cdot F(y, \mathbf{x}) \cdot |\mathbf{x_v}| \Big] - \mathbb{E}\Big[ \ell_\sigma'(|\mathbf{x_w}|) \cdot \mathbb{1}\{\mathbf{x} \notin \mathcal{G}\} \cdot F(y, \mathbf{x}) \cdot |\mathbf{x_v}| \Big]
\end{aligned}
$$

Let $A_1' = \mathbb{E}[\ell_\sigma'(|\mathbf{x_w}|) \cdot \mathbb{1}\{\mathbf{x} \in \mathcal{G}\} \cdot F(y, \mathbf{x}) \cdot |\mathbf{x_v}|]$ and $A_2' = \mathbb{E}[\ell_\sigma'(|\mathbf{x_w}|) \cdot \mathbb{1}\{\mathbf{x} \notin \mathcal{G}\} \cdot F(y, \mathbf{x}) \cdot |\mathbf{x_v}|]$. In the Massart noise case $\mathbb{E}_{y|\mathbf{x}}[F(y, \mathbf{x})] = 1 - 2\eta(\mathbf{x}) \in [1 - 2\eta, 1]$, where $1 - 2\eta > 0$. Therefore, we have that $A_1' \geq (1 - 2\eta) \cdot \mathbb{E}[\ell_\sigma'(|\mathbf{x_w}|) \cdot \mathbb{1}\{\mathbf{x} \in \mathcal{G}\} \cdot |\mathbf{x_v}|]$. When the noise is adversarial, we have $A_1' \geq \mathbb{E}[\ell_\sigma'(|\mathbf{x_w}|) \cdot \mathbb{1}\{\mathbf{x} \in \mathcal{G}\} \cdot |\mathbf{x_v}|] - 2\mathbb{E}[\ell_\sigma'(|\mathbf{x_w}|) \cdot \mathbb{1}\{\mathbf{x} \in \mathcal{G}\} \cdot \mathbb{1}\{y \neq \text{sign}(\langle \mathbf{w}^*, \mathbf{x} \rangle)\} \cdot |\mathbf{x_v}|]$.

For any $\alpha \geq \frac{\sigma}{2\tan\theta}$, we have that

$$
\begin{aligned}
\mathbb{E}\Big[ \ell_\sigma'(|\mathbf{x_w}|) \cdot \mathbb{1}\{\mathbf{x} \in \mathcal{G}\} \cdot |\mathbf{x_v}| \Big] &\geq \mathbb{E}\Big[ \ell_\sigma'(|\mathbf{x_w}|) \cdot \mathbb{1}\{\mathbf{x} \in \mathcal{G}\} \cdot \mathbb{1}_{\{|\mathbf{x_w}| \leq \frac{\sigma}{6}\}} \cdot |\mathbf{x_v}| \Big] \\
&\qquad \text{(since terms are positive)} \\
&\geq \mathbb{E}\left[ \frac{1}{\sigma} \cdot \mathbb{1}\{\mathbf{x} \in \mathcal{G}\} \cdot \mathbb{1}\left\{|\mathbf{x_w}| \leq \frac{\sigma}{6}\right\} \cdot |\mathbf{x_v}| \right] \\
&\qquad \text{(by Proposition A.4)} \\
&\geq \frac{\alpha}{\sigma} \cdot \mathbb{E}\Big[ \mathbb{1}\{\mathbf{x} \in \mathcal{G}\} \cdot \mathbb{1}_{\{|\mathbf{x_w}| \leq \frac{\sigma}{6}\}} \cdot \mathbb{1}_{\{|\mathbf{x_v}| \geq \alpha\}} \Big] \\
&\geq \frac{\alpha}{\sigma} \cdot \mathbb{E}\Big[ \mathbb{1}_{\{|\mathbf{x_w}| \leq \frac{\sigma}{6}\}} \cdot \mathbb{1}_{\{|\mathbf{x_v}| \geq \alpha\}} \Big] \qquad \text{(see Figure 1)} \\
&= \frac{\alpha}{\sigma} \cdot \mathbb{P}\left[ |\mathbf{x_w}| \leq \frac{\sigma}{6} \text{ and } |\mathbf{x_v}| \geq \alpha \right] \overset{\text{def}}{=} A_1
\end{aligned}
$$

Moreover, for some universal constant $C' > 0$, we similarly have

$$\mathbb{E}\left[\ell'_\sigma(|\mathbf{x_w}|) \cdot \mathbb{1}\{\mathbf{x} \notin \mathcal{G}\} \cdot F(y, \mathbf{x}) \cdot |\mathbf{x_v}|\right] \leq \mathbb{E}\left[\ell'_\sigma(|\mathbf{x_w}|) \cdot \mathbb{1}\{\mathbf{x} \notin \mathcal{G}\} \cdot |\mathbf{x_v}|\right] \quad (\text{since } F(y, \mathbf{x}) \leq 1)$$

$$\leq \mathbb{E}\left[\frac{C'}{\sigma} \cdot \mathbb{1}_{\{|\mathbf{x_w}| \leq \frac{\sigma}{2}\}} \cdot \mathbb{1}\{\mathbf{x} \notin \mathcal{G}\} \cdot |\mathbf{x_v}|\right]$$
$$\text{(by Proposition A.4)}$$

$$\leq \frac{C'}{\sigma} \cdot \mathbb{E}\left[\mathbb{1}_{\{|\mathbf{x_w}| \leq \frac{\sigma}{2}\}} \cdot \mathbb{1}_{\{|\mathbf{x_v}| \leq \frac{\sigma}{2\tan\theta}\}} \cdot |\mathbf{x_v}|\right]$$
$$\text{(see Figure 1)}$$

$$\leq \frac{C'}{2 \cdot \tan\theta} \cdot \mathbb{E}\left[\mathbb{1}_{\{|\mathbf{x_w}| \leq \frac{\sigma}{2}\}} \cdot \mathbb{1}_{\{|\mathbf{x_v}| \leq \frac{\sigma}{2\tan\theta}\}}\right]$$

$$\leq \frac{C'}{2 \cdot \tan\theta} \cdot \mathbb{P}\left[|\mathbf{x_w}| \leq \frac{\sigma}{2}\right] \stackrel{\text{def}}{=} A_2$$

Hence, we have shown that, in the Massart noise case, we have $\|\nabla_\mathbf{w}\mathcal{L}_\sigma(\mathbf{w})\|_2 \geq (1 - 2\eta)A_1 - A_2$ as desired. For the adversarial noise case, it remains to bound the following quantity

$$2\,\mathbb{E}\left[\ell'_\sigma(|\mathbf{x_w}|) \cdot \mathbb{1}_{\{\mathbf{x} \in \mathcal{G}\}} \cdot \mathbb{1}_{\{y \neq \text{sign}(\langle \mathbf{w}^*, \mathbf{x}\rangle)\}} \cdot |\mathbf{x_v}|\right] \leq$$

$$\leq \frac{2C'}{\sigma} \cdot \mathbb{E}\left[\mathbb{1}_{\{\mathbf{x} \in \mathcal{G}\}} \cdot \mathbb{1}_{\{|\mathbf{x_w}| \leq \frac{\sigma}{2}\}} \cdot \mathbb{1}_{\{y \neq \text{sign}(\langle \mathbf{w}^*, \mathbf{x}\rangle)\}} \cdot |\mathbf{x_v}|\right]$$

$$\leq \frac{2C'}{\sigma} \cdot \mathbb{E}\left[\mathbb{1}_{\{|\mathbf{x_w}| \leq \frac{\sigma}{2}\}} \cdot \mathbb{1}_{\{y \neq \text{sign}(\langle \mathbf{w}^*, \mathbf{x}\rangle)\}} \cdot |\mathbf{x_v}|\right]$$

$$\leq \frac{2C'}{\sigma} \cdot \sqrt{\text{opt}} \cdot \sqrt{\mathbb{E}\left[|\mathbf{x_v}|^2 \cdot \mathbb{1}_{\{|\mathbf{x_w}| \leq \frac{\sigma}{2}\}}\right]} \stackrel{\text{def}}{=} A_3$$

where the final inequality follows from Cauchy-Schwarz inequality. $\qquad\square$

## C.2  Proof of Lemma 4.3

We restate Lemma 4.3 here for convenience.

**Lemma C.2** (Universally Testable Structure of Surrogate Loss). *Let $D_{\mathcal{X}\mathcal{Y}}$ be any distribution over $\mathbb{R}^d \times \{\pm 1\}$. Consider $\mathcal{L}_\sigma$ as in Equation (4.1). Then, there is a universal constant $C > 0$ and a tester that given a unit vector $\mathbf{w} \in \mathbb{R}^d$, $\delta \in (0, 1)$, $\eta < 1/2$, $\gamma > 0$, $\lambda \geq 1$, $\sigma \leq \frac{1}{C\lambda^C}$ and a set $S$ of i.i.d. samples from $D_{\mathcal{X}\mathcal{Y}}$ with size at least $C \cdot \frac{d^4}{\sigma^2\delta}\log(d)\lambda^C$, runs in time $\text{poly}(d, \lambda, \frac{1}{\sigma}, \frac{1}{\delta}, \log(\frac{1}{\gamma}))$ and satisfies the following specifications*

(a) *If the tester accepts $S$, then, the following statements are true for the minimum error $\text{opt}_S$ achieved by some origin-centered halfspace on $S$ and the optimum vector $\mathbf{w}_S^* \in \mathbb{S}^{d-1}$*

- *If the noise is Massart with associated rate $\eta$ and $\|\nabla_\mathbf{w}\mathcal{L}_\sigma(\mathbf{w}; S)\|_2 \leq \frac{1-2\eta}{C\lambda^C\gamma^4}$ then either $\measuredangle(\mathbf{w}, \mathbf{w}_S^*) \leq \frac{C\lambda^C(1+\gamma^4)}{1-2\eta} \cdot \sigma$ or $\measuredangle(-\mathbf{w}, \mathbf{w}_S^*) \leq \frac{C\lambda^C(1+\gamma^4)}{1-2\eta} \cdot \sigma$.*

- *If the noise is adversarial with $\text{opt}_S \leq \frac{\sigma}{C\lambda^C}$ and $\|\nabla_\mathbf{w}\mathcal{L}_\sigma(\mathbf{w}; S)\|_2 < \frac{1}{C\lambda^C\gamma^4}$ then either $\measuredangle(\mathbf{w}, \mathbf{w}_S^*) \leq C\lambda^C(1+\gamma^4) \cdot \sigma$ or $\measuredangle(-\mathbf{w}, \mathbf{w}_S^*) \leq C\lambda^C(1+\gamma^4) \cdot \sigma$.*

(b) *If the marginal $D_{\mathcal{X}}$ is $\lambda$-nice and $\gamma$-Poincaré, then the tester accepts $S$ with probability at least $1 - \delta$.*

*Proof of Lemma 4.3.* The testing algorithm receives $\mathbf{w} \in \mathbb{S}^{d-1}$, $\delta \in (0, 1)$, $\eta < 1/2$, $\gamma > 0$, $\lambda \geq 1$, $\sigma \leq \frac{1}{2\lambda}$ and a set $S \subset \mathbb{R}^d \times \{\pm 1\}$ and does the following for some sufficiently large $C_1 > 0$

1. If $\mathbb{P}_{(\mathbf{x}, y) \in S}[|\langle \mathbf{w}, \mathbf{x}\rangle| \leq \frac{\sigma}{6}] \leq \frac{\sigma}{C_1\lambda^{C_1}}$ or $\mathbb{P}_{(\mathbf{x}, y) \in S}[|\langle \mathbf{w}, \mathbf{x}\rangle| \leq \frac{\sigma}{2}] > \sigma \cdot C_1\lambda^{C_1}$, then **reject**.

2. Compute the $(d-1) \times (d-1)$ matrices $M_S^+$ and $M_S^-$ as follows:

$$M_S^+ = \mathop{\mathbb{E}}_{(\mathbf{x},y) \in S} \left[ (\mathrm{proj}_{\perp \mathbf{w}} \mathbf{x})(\mathrm{proj}_{\perp \mathbf{w}} \mathbf{x})^T \cdot \mathbb{1}_{\{|\langle \mathbf{w}, \mathbf{x} \rangle| \leq \frac{\sigma}{2}\}} \right]$$

$$M_S^- = \mathop{\mathbb{E}}_{(\mathbf{x},y) \in S} \left[ (\mathrm{proj}_{\perp \mathbf{w}} \mathbf{x})(\mathrm{proj}_{\perp \mathbf{w}} \mathbf{x})^T \cdot \mathbb{1}_{\{|\langle \mathbf{w}, \mathbf{x} \rangle| \leq \frac{\sigma}{6}\}} \right]$$

3. Run the (maximum singular value) spectral tester of Proposition A.2 on $M_S^+$ given $\delta \leftarrow \frac{\delta}{4}$, $\lambda \leftarrow C_1 \lambda^{C_1}$ and $\theta \leftarrow \frac{C_1 \sigma \lambda^{C_1}}{2}$, i.e., **reject** if the maximum singular value of $M_S^+$ is greater than $\sigma \cdot C_1 \lambda^{C_1}$.

4. Run the (minimum singular value) spectral tester of Proposition A.2 on $M_S^-$ given $\delta \leftarrow \frac{\delta}{4}$, $\lambda \leftarrow C_1 \lambda^{C_1}$ and $\theta \leftarrow \frac{2\sigma}{C_1 \lambda^{C_1}}$, i.e., **reject** if the minimum singular value of $M_S^-$ is less than $\frac{\sigma}{C_1 \lambda^{C_1}}$.

5. Run the hypercontractivity tester on $S' = \{\mathrm{proj}_{\perp \mathbf{w}} \mathbf{x} : (\mathbf{x}, y) \in S \text{ and } |\langle \mathbf{w}, \mathbf{x} \rangle| \leq \sigma\}$, i.e., solve an appropriate SDP (see Prop. 3.5 with $\gamma \leftarrow \gamma, \delta \leftarrow \delta/4$) and **reject** if the solution is larger than a specified threshold. Otherwise, **accept**.

For part (a), we suppose that the testing algorithm has accepted $S$. Therefore, $S$ has passed all the tests required for part (a) of Lemma 3.2 and there exists a universal constant $C' > 0$ such that

$$\mathop{\mathbb{P}}_{(\mathbf{x},y) \in S} \left[ |\langle \mathbf{v}, \mathbf{x} \rangle| \geq \frac{1}{C' \lambda^{C'}} \,\Big|\, |\langle \mathbf{w}, \mathbf{x} \rangle| \leq \sigma \right] \geq \frac{1}{C' \lambda^{C'} \gamma^4}$$

Moreover, we have $\frac{\sigma}{C' \lambda^{C'}} < \mathbb{P}_{(\mathbf{x},y) \in S}[|\langle \mathbf{w}, \mathbf{x} \rangle| \leq \frac{\sigma}{6}] \leq \mathbb{P}_{(\mathbf{x},y) \in S}[|\langle \mathbf{w}, \mathbf{x} \rangle| \leq \frac{\sigma}{2}] < \sigma \cdot C' \lambda^{C'}$ and

$$\mathop{\mathbb{E}}_{(\mathbf{x},y) \in S} \left[ \langle \mathbf{v}, \mathbf{x} \rangle^2 \,\Big|\, |\langle \mathbf{w}, \mathbf{x} \rangle| \leq \frac{\sigma}{2} \right] \leq C' \lambda^{C'}$$

$$\mathop{\mathbb{E}}_{(\mathbf{x},y) \in S} \left[ \langle \mathbf{v}, \mathbf{x} \rangle^2 \,\Big|\, |\langle \mathbf{w}, \mathbf{x} \rangle| \leq \frac{\sigma}{6} \right] \geq \frac{1}{C' \lambda^{C'}}$$

Since Proposition 4.2 holds for any distribution, it will also hold for the empirical distribution (uniform on $S$). We apply Proposition 4.2 with $\alpha = \frac{1}{C' \lambda^{C'}}$ to lower bound $\|\nabla_\mathbf{w} \mathcal{L}_\sigma(\mathbf{w}; S)\|_2$ (or $\|\nabla_\mathbf{w} \mathcal{L}_\sigma(-\mathbf{w}; S)\|_2$) as follows

$$\|\nabla_\mathbf{w} \mathcal{L}_\sigma(\mathbf{w}; S)\|_2 \geq A_1(\alpha) - A_2 - A_3 \qquad \text{(adversarial noise case)}$$
$$\|\nabla_\mathbf{w} \mathcal{L}_\sigma(\mathbf{w}; S)\|_2 \geq (1 - 2\eta) \cdot A_1(\alpha) - A_2 \qquad \text{(Massart noise case)}$$

Combining the above inequalities with the bounds implied by the fact that $S$ has passed the tests, concludes the proof of part (a), since (after observing that $\tan \theta \geq \theta$) we get

$$\|\nabla_\mathbf{w} \mathcal{L}_\sigma(\mathbf{w}; S)\|_2 \geq \frac{3}{C \lambda^C \gamma^4} - \frac{\sqrt{C} \sigma \lambda^{C/2}}{\theta} - \sqrt{\frac{\mathrm{opt} \cdot C \cdot \lambda^C}{\sigma}} \qquad \text{(adversarial noise case)}$$

$$\|\nabla_\mathbf{w} \mathcal{L}_\sigma(\mathbf{w}; S)\|_2 \geq \frac{3(1 - \eta)}{C \lambda^C \gamma^4} - \frac{\sqrt{C} \sigma \lambda^{C/2}}{\theta} \qquad \text{(Massart noise case)}$$

For part (b), we follow a similar recipe as the one used to prove part (b) of Lemma 3.2, i.e., we use the following reasoning to show that the tests will pass with probability at least $1 - \delta$

1. We assume that the marginal distribution $D_\mathcal{X}$ is $\lambda$-nice and $\gamma$-Poincaré.

2. We use Proposition A.3 to bound the values of the tested quantities under the true distribution.

3. We use appropriate concentration results (Hoeffding/Chernoff Bounds and Proposition A.2) to show that, since $|S|$ is large enough, each of the empirical quantities at hand does not deviate a lot from its mean.

This concludes the proof of Lemma 4.3. $\qquad \square$

## C.3 Proof of Main Theorem

We restate the main Theorem here for convenience.

**Theorem C.3** (Efficient Universal Tester-Learner for Halfspaces). *Let $D_{\mathcal{X}\mathcal{Y}}$ be any distribution over $\mathbb{R}^d \times \{\pm 1\}$. Let $\mathcal{C}$ be the class of origin centered halfspaces in $\mathbb{R}^d$. Then, for any $\lambda \geq 1$, $\gamma > 0$, $\epsilon > 0$ and $\delta \in (0,1)$, there exists an universal tester-learner for $\mathcal{C}$ w.r.t. the class of $\lambda$-nice and $\gamma$-Poincaré marginals up to error $\mathrm{poly}(\lambda) \cdot (1+\gamma^4) \cdot \mathsf{opt} + \epsilon$, where $\mathsf{opt} = \min_{\mathbf{w}\in\mathbb{S}^{d-1}} \mathbb{P}_{D_{\mathcal{X}\mathcal{Y}}}[y \neq \mathrm{sign}(\langle\mathbf{w},\mathbf{x}\rangle)]$, and error probability at most $\delta$, using a number of samples and running time $\mathrm{poly}(d,\lambda,\gamma,\frac{1}{\epsilon},\log\frac{1}{\delta})$.*

*Moreover, if the noise is Massart with given rate $\eta < 1/2$, then the algorithm achieves error $\mathsf{opt} + \epsilon$ with time and sample complexity $\mathrm{poly}(d,\lambda,\gamma,\frac{1}{\epsilon},\frac{1}{1-2\eta},\log\frac{1}{\delta})$.*

*Proof of Theorem 4.1.* Note that we will give the algorithm for $\delta \leftarrow \delta' = 1/3$ since we can reduce the probability of failure with repetition (repeat $O(\log\frac{1}{\delta})$ times, accept if the rate of acceptance is $\Omega(1)$ and output the halfspace achieving the minimum test error among the halfspaces returned).

For reader's convenience, we now restate the algorithm on page 9 (note that together with the algorithm we include additional detail relevant to the analysis). The algorithm receives $\lambda \geq 1$, $\gamma > 0$, $\epsilon > 0$ and $\eta \in (0,1/2) \cup \{1\}$ (say $\eta = 1$ when we are in the agnostic case) and does the following for some appropriately large universal constant $C_1, C_2 > 0$.

1. First, create a set of parameters $\Sigma$ and parameters $E = \frac{\epsilon}{C_1\lambda^{C_1}}$ and $A > 0$ as follows. If $\eta = 1$, then $\Sigma$ is an $\frac{E}{C_1\lambda^{C_1}}$-cover of the interval $\left[0, \frac{1}{C_1\lambda^{C_1}}\right]$ and $A = \frac{1}{C_1\lambda^{C_1}\gamma^4}$. Otherwise, let $\Sigma = \left\{\frac{E\cdot(1-2\eta)}{C_1\lambda^{C_1}(1+\gamma^4)}\right\}$ and $A = \frac{1-2\eta}{C_1\lambda^{C_1}\gamma^4}$.

2. Then, draw a set $S_1$ of $C_2\left(\frac{\lambda d}{\gamma\epsilon}\right)^{C_2}$ i.i.d. samples from $D_{\mathcal{X}\mathcal{Y}}$ and run PSGD, as specified in Proposition A.5 with $\epsilon \leftarrow A$, $\delta \leftarrow \frac{\delta'}{C_1}$ on the loss $\mathcal{L}_\sigma$ for each $\sigma \in \Sigma$.

3. Form a list $L$ with all the pairs of the form $(\mathbf{w}, \sigma)$ where $\mathbf{w} \in \mathbb{S}^{d-1}$ is some iterate of the PSGD subroutine performed on $\mathcal{L}_\sigma$.

4. Draw a fresh set $S_2$ of $C_2\left(\frac{\lambda d}{\gamma\epsilon}\right)^{C_2}$ i.i.d. samples from $D_{\mathcal{X}\mathcal{Y}}$ and compute for each $(\mathbf{w}, \sigma) \in L$ the value $\|\nabla_\mathbf{w}\mathcal{L}_\sigma(\mathbf{w}; S_2)\|_2$. If, for some $\sigma \in \Sigma$, $\|\nabla_\mathbf{w}\mathcal{L}_\sigma(\mathbf{w}; S_2)\|_2 > A$ for all $(\mathbf{w}, \sigma) \in L$, then **reject**.

5. Update $L$ by keeping for each $\sigma \in \Sigma$ only one pair of the form $(\mathbf{w}, \sigma)$ for which we have $\|\nabla_\mathbf{w}\mathcal{L}_\sigma(\mathbf{w}; S_2)\|_2 \leq A$.

6. Run the following tests for each $(\mathbf{w}, \sigma) \in L$ to ensure that part (a) of Lemma 4.3 holds for each of the elements of $L$, i.e., that any stationary point of the surrogate loss that lies in $L$ is angularly close to the empirical risk minimizer (or the same holds for the inverse vector).

   - If $\mathbb{P}_{(\mathbf{x},y)\in S_2}[|\langle\mathbf{w},\mathbf{x}\rangle| \leq \frac{\sigma}{6}] \leq \frac{\sigma}{C_1\lambda^{C_1}}$ or $\mathbb{P}_{(\mathbf{x},y)\in S_2}[|\langle\mathbf{w},\mathbf{x}\rangle| \leq \frac{\sigma}{2}] > \sigma \cdot C_1\lambda^{C_1}$, then **reject**.
   - Compute the $(d-1) \times (d-1)$ matrices $M_{S_2}^+$ and $M_{S_2}^-$ as follows:
   $$M_{S_2}^+ = \mathop{\mathbb{E}}_{(\mathbf{x},y)\in S_2}\left[(\mathrm{proj}_{\perp\mathbf{w}}\mathbf{x})(\mathrm{proj}_{\perp\mathbf{w}}\mathbf{x})^T \cdot \mathbb{1}_{\{|\langle\mathbf{w},\mathbf{x}\rangle|\leq\frac{\sigma}{2}\}}\right]$$
   $$M_{S_2}^- = \mathop{\mathbb{E}}_{(\mathbf{x},y)\in S_2}\left[(\mathrm{proj}_{\perp\mathbf{w}}\mathbf{x})(\mathrm{proj}_{\perp\mathbf{w}}\mathbf{x})^T \cdot \mathbb{1}_{\{|\langle\mathbf{w},\mathbf{x}\rangle|\leq\frac{\sigma}{6}\}}\right]$$
   - Run the (maximum singular value) spectral tester of Proposition A.2 on $M_{S_2}^+$ given $\delta \leftarrow \frac{\delta'}{C_1}$, $\lambda \leftarrow C_1\lambda^{C_1}$ and $\theta \leftarrow \frac{C_1\sigma\lambda^{C_1}}{2}$, i.e., **reject** if the maximum singular value of $M_{S_2}^+$ is greater than $\sigma \cdot C_1\lambda^{C_1}$.
   - Run the (minimum singular value) spectral tester of Proposition A.2 on $M_{S_2}^-$ given $\delta \leftarrow \frac{\delta'}{C_1}$, $\lambda \leftarrow C_1\lambda^{C_1}$ and $\theta \leftarrow \frac{2\sigma}{C_1\lambda^{C_1}}$, i.e., **reject** if the minimum singular value of $M_{S_2}^-$ is less than $\frac{\sigma}{C_1\lambda^{C_1}}$.

- Run the hypercontractivity tester on $S' = \{\text{proj}_{\perp \mathbf{w}} \mathbf{x} : (\mathbf{x}, y) \in S_2 \text{ and } |\langle \mathbf{w}, \mathbf{x} \rangle| \leq \sigma\}$, i.e., solve an appropriate SDP (see Prop. 3.5 with $\gamma \leftarrow \gamma$, $\delta \leftarrow \delta'/C_1$) and **reject** if the solution is larger than a specified threshold.

7. Run the following tests for each pair of the form $(\mathbf{w}, \sigma)$ and $(-\mathbf{w}, \sigma)$ where $(\mathbf{w}, \sigma) \in L$ to ensure that part (a) of Lemma 3.1 is activated, i.e., that the distance of a vector from the empirical risk minimizer is an accurate proxy for the error of the corresponding halfspace. Set $\theta(\sigma) = \frac{(1+\gamma^4)\sigma}{A\gamma^4}$.

    - If $\mathbb{P}_{(\mathbf{x},y) \in S_2}[|\langle \mathbf{w}, \mathbf{x} \rangle| \leq \theta] > C_1 \lambda^{C_1} \theta$ then **reject**.
    - Compute the $(d-1) \times (d-1)$ matrix $M_{S_2}$ as follows[6]:

$$M_{S_2} = \mathop{\mathbb{E}}_{(\mathbf{x},y) \in S_2}\left[\sum_{i=2}^{\infty} \frac{(\text{proj}_{\perp \mathbf{w}} \mathbf{x})(\text{proj}_{\perp \mathbf{w}} \mathbf{x})^T}{(i-1)^2} \mathbb{1}\{|\langle \mathbf{w}, \mathbf{x} \rangle| \in [(i-1)\theta, i\theta)\}\right]$$

    - Run the spectral tester of Proposition A.2 on $M_S$ given $\delta \leftarrow \frac{\delta'}{C_1}$, $\lambda \leftarrow C_1 \lambda^{C_1}$ and $\theta \leftarrow \frac{C_1}{2} \theta \lambda^{C_1}$, i.e., compute $\|M_S\|_{\text{op}}$ and if $\|M_S\|_{\text{op}} > C_1 \theta \lambda^{C_1}$, then **reject**.

8. Otherwise, **accept** and output the vector $\mathbf{w}$ that achieves the smallest empirical error on $S_2$ among the vectors in the list $L$.

For the following, let $\alpha = 1$ in the Massart noise case and $\alpha = C_1 \lambda^{C_1} \gamma^4$ in the adversarial noise case. Consider also $\text{opt}_{S_2}$ to be the error of the origin-centered halfspace with the minimum empirical error on $S_2$ and $\mathbf{w}_{S_2}^*$ the corresponding optimum vector.

**Soundness.** We first prove the soundness condition, i.e., that the following implication holds with probability at least $1 - \delta'$ over the samples:

$$\text{If the tester accepts, then } \mathop{\mathbb{P}}_{D_{\mathcal{X}\mathcal{Y}}}[y \neq \text{sign}(\langle \mathbf{w}, \mathbf{x} \rangle)] \leq \alpha \cdot \text{opt} + \epsilon$$

The tester accepts only if for every $\sigma \in \Sigma$, we have some $\mathbf{w} \in L$ with $\|\nabla_{\mathbf{w}} \mathcal{L}_\sigma(\mathbf{w}; S_2)\|_2 \leq A$ (step 4) and for which part (a) of each of Lemmas 4.3 (step 6) and 3.1 (step 7) is activated. Therefore, in the Massart noise case, for any $\sigma \in \Sigma$, there is some $\mathbf{w}$ such that either $(\mathbf{w}, \sigma) \in L$ or $(-\mathbf{w}, \sigma) \in L$ and also

$$\angle(\mathbf{w}, \mathbf{w}_{S_2}^*) \leq \frac{1 + \gamma^4}{\gamma^4} \cdot \frac{\sigma}{A} \overset{\text{def}}{=} \theta \tag{C.1}$$

$$\mathop{\mathbb{P}}_{S_2}[y \neq \text{sign}(\langle \mathbf{w}, \mathbf{x} \rangle)] \leq \text{opt}_{S_2} + C' \lambda^{C'} \cdot \theta \tag{C.2}$$

In the adversarial noise case, the above are true conditional on $\sigma$ being such that $\text{opt}_{S_2} \leq \frac{\sigma}{C' \lambda^{C'}}$.

Therefore, in the Massart noise case, the above are true for $\sigma = \frac{E(1-2\eta)}{C_1 \lambda^{C_1}(1+\gamma^4)}$ which gives

$$\mathop{\mathbb{P}}_{S_2}[y \neq \text{sign}(\langle \mathbf{w}, \mathbf{x} \rangle)] \leq \text{opt}_{S_2} + C' \lambda^{C'} E$$

In the agnostic case, condition C.2 is true for some $\sigma \in [0, \frac{1}{C_1 \lambda^{C_1}}]$ such that

$$\frac{\sigma}{C' \lambda^{C'}} - \frac{1}{C_1 \lambda^{C_1}} \leq \text{opt}_{S_2} \leq \frac{\sigma}{C' \lambda^{C'}}$$

unless $\text{opt} > \frac{1}{C_1 C' \lambda^{C_1 + C'}}$, in which case any halfspace has error at most $1 = \text{opt} \cdot (C_1 C' \lambda^{C_1 + C'})$. Hence we get

$$\mathop{\mathbb{P}}_{S_2}[y \neq \text{sign}(\langle \mathbf{w}, \mathbf{x} \rangle)] \leq \text{poly}(\lambda) \cdot (1 + \gamma^4) \cdot \text{opt}_{S_2} + C' \lambda^{C'} E$$

Soundness follows from the fact that if $|S_2|$ is sufficiently large (but still polynomial in every parameter, since the VC dimension of the class of halfspaces in $\mathbb{R}^d$ is $d+1$), then $|\text{opt}_{S_2} - \text{opt}| \leq \frac{\epsilon}{C_1 \lambda^{C_1}(1+\gamma)^4}$ with probability at least $1 - \delta'$.

---

[6]Note that only at most $|S_2|$ many terms below are non-zero, hence $M_{S_2}$ can be computed efficiently.

**Completeness.** Suppose now that the marginal is indeed $\lambda$-nice and $\gamma$-Poincaré. Then, for sufficiently large $S_1$, after step 3, $L$ will contain a stationary point of $\mathcal{L}_\sigma(\,\cdot\,; D_{\mathcal{X}\mathcal{Y}})$ for each $\sigma \in \Sigma$, due to Proposition A.5. If $S_2$ is large enough, then steps 4, 6 and 7 will each accept with probability at least $1 - \delta'/C_1$, due to part (b) of Lemmas 4.3 and 3.1, as well as the fact that each coordinate of $\nabla_{\mathbf{w}} \mathcal{L}_\sigma(\mathbf{w}; S_2)$ has bounded second moment (Proposition A.3) and therefore $\nabla_{\mathbf{w}} \mathcal{L}_\sigma(\mathbf{w}; S_2)$ is concentrated around $\nabla_{\mathbf{w}} \mathcal{L}_\sigma(\mathbf{w}; D_{\mathcal{X}\mathcal{Y}})$ for any fixed $\mathbf{w}$ such that $(\mathbf{w}, \sigma) \in L$ (we also need a union bound over $L$). Hence, in total, the tester will accept with probability at least $1 - \delta'$. $\qquad\square$

