# OpenReview forum: "Tester-Learners for Halfspaces: Universal Algorithms"
_NeurIPS.cc/2023/Conference — NeurIPS 2023 oral_

### Official Review · Reviewer_eYN2 · 2023-07-05

**Soundness:** 3 good
**Presentation:** 3 good
**Contribution:** 3 good
**Rating:** 7
**Confidence:** 3

**Summary:**

This work studies the problem of designing tester-learners for halfspaces in the agnostic setting and under Massart noise. In the setting of tester-learners, we do not make distributional assumptions on our samples, but rather require that our algorithm produces an accurate output whenever the input satisfies a sequence of efficiently computable tests. Further, if the samples actually come from a specific target distribution, it is required that the tests pass with high probability.

Previous tester-learners for noisy halfspaces used tests tailored towards a *specific* distribution, e.g., a specific strongly log-concave distribution. As a consequence, these testers might reject samples from a different strongly log-concave distribution although algorithmically it would have been possible to learn the underlying halfspace. The authors propose a tester which simultaneously accepts for all Poincaré distributions that satisfy some natural niceness conditions. This class includes all isotropic strongly log-concave distributions. When their tester accepts, the tester-learner outputs a halfspace achieving error $O(\mathrm{opt}) + \varepsilon$, where $\mathrm{opt}$ is the error of the optimal halfspace on the distribution of the input. For the Massart setting they can handle all isotropic log-concave distributions and achieve error $\mathrm{opt} + \varepsilon$.

The authors argue that previous works were inherently limited to work with only a single distribution since they were matching moments with this specific distribution. In their work, they overcome this issue by only checking the actual properties they need in the proof of accuracy which are not tied to any specific distribution.

**Strengths:**

In my eyes, designing tester learners which simultaneously accept for a large class of distribution is an important step for bringing the framework of testable learning closer to practice. The motivation for testable learning was that designing algorithms that only work under specific distributional assumptions has little value in practice since likely these assumptions will not be met. Thus, we would at least like to know if we can trust the output of our algorithm when we run it on actual data or not. While previous tester-learners did have this benefit, it seems that the proposed testers "overfit" to specific distributions. However, the chances that real world data comes from any specific distribution are rather small. Thus, providing testers which work for larger classes of distributions is an important step.

On a technical level, their tester has a natural explanation that I like: It simply checks the properties that are need in the proof in the non-testable setting. Further, the paper is very well-written (except the point below).

**Weaknesses:**

I feel that the technical overview that the authors provide might be hard to understand if the reader is not familiar with the arguments in previous work or at least the general area. It would be nice if it could be adjusted slightly to make it more accessible to a wider audience.

Also, specifically for the agnostic setting, it would be useful to compare your approximation factor to (a) testable learners that only work for one specific (or a more restricted class of distributions) and (b) non-testable learners for the same distribution class you consider.

**Questions:**

Related to the above: In the agnostic setting, your error bound scales as $\mathrm{poly}(\lambda) (1+\gamma^4) \mathrm{opt} + \varepsilon$, where $\lambda$ is the niceness parameter of the marginal distribution and $\gamma$ the Poincaré constant. How does this compare to the non-testable learning error?

**Limitations:**

Limitations were addressed appropriately.

---

> ### Author Rebuttal · Authors · 2023-08-09
>
> We wish to thank the anonymous reviewer for their suggestions and comments!
>
> Prior work in the area has demonstrated that, in the distribution specific setting, assuming that the marginals are “nice” (i.e., that $\lambda$ is some constant) is sufficient and there is no dependence of the error bound on the Poincare parameter (hence, the error bound would be in that case $\mathrm{poly}(\lambda)\mathrm{opt} + \epsilon$). We note that the precise dependence on the niceness parameter $\lambda$ was not stated explicitly in prior work. As for the dependence on the Poincare parameter, it arises due to the need to provide efficiently testable bounds to certain quantities (see Lemma 3.2) which, in the non-testable setting, admit tighter bounds; in other words, the bounds we use involve some slackness due to testability requirements.
>
>   Regarding prior work on regular (non universal) testable learning, prior work only provides results with respect to strongly log-concave marginals (based on the technology of fooling halfspaces via moment matching), where $\gamma$ is always a constant (see also lines 200-206).

---

> > ### Comment · Reviewer_eYN2 · 2023-08-14
> >
> > Thank you for your comments and clarification. I believe it would be useful to include the first one in the submission.

---

### Official Review · Reviewer_EFhK · 2023-07-05

**Soundness:** 3 good
**Presentation:** 3 good
**Contribution:** 2 fair
**Rating:** 6
**Confidence:** 3

**Summary:**

The paper develops a tester-learner for halfspaces in the recently proposed testable learning framework of [RV23]. A recent work [GKSV23] had studied the same problem and proposed tester-learners for halfspaces with Massart noise and in the agnostic setting, for any particular strongly-log concave distribution. This work qualitatively improves upon [GKSV23] by developing a tester-learner for a class of strongly log-concave distributions rather than a fixed strongly log-concave distribution, and also provides better algorithmic complexity for the agnostic case. In fact, this work captures $\gamma$-Poincare distributions which contain strongly log-concave distributions and under the [KLS95] conjecture also all log-concave distributions. In the Massart noise case, their algorithm works for log-concave distributions unconditionally.
 While following the recently developed approach (and used by [GKSV23]) of non-convex SGD for learning halfspaces over certain good distributions, this paper does a more careful analysis of the gradient lower bound leading to the requirement of a certain hypercontractivity property (which holds for Poincare distributions). The paper uses techniques for previous work [KS17] to show that sum-of-squares (SoS) SDP can be used to certify this hypercontractivity which is a main technical novelty.


**Strengths:**

The paper proves better and qualitatively more general and stronger bounds for testable learning of halfspaces over a general class of distributions. The technical contributions - the use of properties of Poincare distributions and using SoS to certify hypercontractivity -  are interesting and novel in the context of this problem, and provide for universal tester-learners. The paper is well written and effectively conveys its main contributions and the techniques used. Overall, the paper is technically solid.

**Weaknesses:**

The paper follows the same roadmap of [GKSV23] with better analysis and a new tester (based on previous work [KS17]) for hypercontractivity, and in that sense feels somewhat incremental. The results are also moderate improvements and generalizations of previous results and not entirely unexpected.

**Questions:**

1. It may be helpful to clarify why the main Theorem 4.1 also captures log-concave distributions unconditionally for the case with Massart noise, while only capturing strongly log-concave distributions in the agnostic setting.

2. Is it clear that the previous work of [GKSV23] cannot be tweaked (e.g. by discretizing the space of the vector of moments to be matched) to work for the class of strongly log-concave distributions?


**Limitations:**

Yes

---

> ### Author Rebuttal · Authors · 2023-08-09
>
> We thank the reviewer for their constructive feedback. Here are our responses to the questions, which we will aim to incorporate in a future revision:
>
> Theorem 4.1 states that in the Massart noise case, we can achieve the optimal $\mathrm{opt}+\epsilon$ guarantee, while in the agnostic setting, we get an approximately optimal guarantee which is proportional to a polynomial of the Poincare parameter ($\gamma$) of the target class of marginal distributions. Therefore, in the agnostic setting, we obtain a constant approximation factor only when $\gamma$ is constant (which is true for strongly log-concave distributions and conditionally true for log-concave distributions). In the Massart noise case, however, even when $\gamma$ is polynomial in the dimension, the guarantee is optimal (and the runtime is still polynomial in the dimension). It is known that the parameter $\gamma$ that corresponds to isotropic log-concave distributions is indeed bounded by a polynomial in the ambient dimension (see, e.g., [1]). We will add such a discussion in the final version of the paper.
>
> It is not clear whether there is a more direct approach to providing results in the universal setting based on prior work. Recall that (roughly) the work of [GKSV23] considers the distribution of examples conditioned to a number of strips perpendicular to vector w and compares the moments of each of these conditional distributions to what these moments should be (under a specific distribution). Now, even for a single one of these strips, it is unknown how to determine whether a set of moments indeed matches those that would arise from a strongly log-concave distribution (which is not given to us in advance).
>
> Even the seemingly simpler task of determining whether a set of low-degree moments of a distribution matches those of some unknown strongly log-concave distribution is not known to be achievable efficiently (using a method employing some form of discretization, or any other method). Indeed, such an algorithm would in particular give a novel alternative method for certifying hypercontractivity of an unknown strongly log-concave distribution, which is a task that is only known to be achievable using a highly sophisticated sum-of-squares approach [3].
> However, the moment discretization approach might conceivably be useful towards generalizing the results of [2] in the universal setting.
>
> [1] Chen, Y. (2021). An almost constant lower bound of the isoperimetric coefficient in the KLS conjecture. Geometric and Functional Analysis, 31, 34-61.
>
> [2] Gollakota, A., Klivans, A. R., & Kothari, P. K. (2023). A moment-matching approach to testable learning and a new characterization of rademacher complexity. STOC 2023.
>
> [3] Pravesh K Kothari and Jacob Steinhardt. Better agnostic clustering via relaxed tensor
> norms. arXiv preprint arXiv:1711.07465, 2017.

---

> > ### Comment · Reviewer_EFhK · 2023-08-15
> >
> > I thank the authors for their clarifications. However, my moderate concerns (listed in the Weaknesses) remain, and I will retain my rating.

---

### Official Review · Reviewer_9R84 · 2023-07-07

**Soundness:** 4 excellent
**Presentation:** 4 excellent
**Contribution:** 4 excellent
**Rating:** 8
**Confidence:** 3

**Summary:**

The paper proposes the first tester-learner for learning the class of halfspaces universally over a class of strongly log-concave distributions (or the class of all log-concave distributions under the KLS assumption). Unlike prior works that crucially rely on testing a specific given marginal distribution while rejecting other well-behaved distributions, the proposed algorithm can accept the marginal distribution as long as it is in the family $\mathcal{D}$. The proposed algorithm is motivated by the tester-learner using non-convex SGD [GKSV23] but with a more careful analysis on the anti-concentration property. The algorithm is powered by the SOS program to check the hypercontractivity of the desired random variable $Z$ that exhibits certain anti-concentration properties, which is known suffices for identifying the Poincare distributions ([KS17]).


**Strengths:**

Designing provable tester-learners addresses the issue that traditional learning algorithms crucially rely on distribution assumptions which could be stringent in practice, which is of importance both from the practical and theoretical sides. This paper proposes the first tester-learner that universally accepts any distribution from a broad family of distributions that satisfy a Poincare inequality, largely generalizing those algorithms that only accepts a certain distribution. The technical contribution lies in identifying the Poincare distributions by applying SoS framework to certify a hypercontractive property of the constructed random variable, which generalizes the known results for standard Gaussian to distributions with weak anti-concentration properties. The technique could be of independent interest for testable weak anti-concentration.


**Weaknesses:**

From the technical side, the SoS program for checking the hypercontractive already exists in [KS17]. The contrition of this paper is more on the application side of such a framework on the problem of testing-learning halfspaces with additional assumptions of certain concentration and anti-concentration. That being said, it is a novel application of such a framework given that it solves a very interesting and important problem.


**Questions:**

While Massart noise is considered in this paper, is it possible to directly apply the techniques to other more challenging types of noise?

**Limitations:**

No concerns.

---

> ### Author Rebuttal · Authors · 2023-08-09
>
> We wish to thank the anonymous reviewer for their feedback and for appreciating our work!
>
> The reviewer is right that it is an interesting open question whether our techniques can be applied to achieve optimal guarantees (i.e., $\mathrm{opt}+\epsilon$) for testably learning halfspaces when the noise model is more challenging than Massart. We believe that such results could conceivably be accomplished by future work for other types of noise, e.g., for Tsybakov noise (by adapting Proposition 4.2). But as the reviewer is likely aware, at least under adversarial noise (where we achieve $O(\mathrm{opt})+\epsilon$), there is evidence (e.g. see [1], [2], [3], [4]) that achieving optimal guarantees in polynomial time is impossible.
>
>
> [1] Diakonikolas, I., Kane, D.M., & Zarifis, N. (2020). Near-Optimal SQ Lower Bounds for Agnostically Learning Halfspaces and ReLUs under Gaussian Marginals. NeurIPS 2020.
>
> [2] Goel, S., Gollakota, A., & Klivans, A.R. (2020). Statistical-Query Lower Bounds via Functional Gradients. NeurIPS 2020.
>
> [3] Diakonikolas, I., Kane D.M., & Ren, L. (2023). Near-Optimal Cryptographic Hardness of Agnostically Learning Halfspaces and ReLU Regression under Gaussian Marginals. ICML 2023.
>
> [4] Tiegel, S. (2023). Hardness of Agnostically Learning Halfspaces from Worst-Case Lattice Problems. COLT 2023.

---

> > ### Comment · Reviewer_9R84 · 2023-08-18
> > **Official Comment by Reviewer 9R84**
> >
> > I thank the authors for their insightful feedback. My reviews and score remain the same.

---

### Official Review · Reviewer_AEBD · 2023-07-07

**Soundness:** 4 excellent
**Presentation:** 3 good
**Contribution:** 4 excellent
**Rating:** 8
**Confidence:** 4

**Summary:**

Learning halfspaces is a very well studied problem in machine learning. In the agnostic (adversarial label noise) and distribution free setting, this problem has been known to be computationally intractable. As a result, there have been several works of agnostic learning in distribution specific settings (where the marginal distribution belongs to a particular family of distributions, say Gaussian or log-concave). In this scenario, the learner has an error of the form $OPT + \epsilon$, where OPT denotes the optimal 0-1 error. This however has complexity $d^{1/\epsilon^2}$, and the exponential dependency on $1/\epsilon$ is tight. This motivates the designing of learning algorithms that have better sample complexity with respect to $1/\epsilon$, whereas the error becomes $f(OPT) + \epsilon$ for some function f.

Often these works use a single distribution as the target marginal distribution. In this work, the authors studied this problem with respect to a set of marginal distributions (Universal testable learning, Definition 1.1). The authors studied this problem in the newly introduced Testable learning framework  by Rubinfeld and Vasilyan. Here the goal is if the tester accepts, then with high probability the output of the learning is close to some function of OPT, and if the data satisfies the distributional assumptions, the algorithm accepts with high probability.

Here the authors design a universal tester learner for Halfspaces with respect to distributions with bounded Poincare constant (Definition 2.4) and concentration and anti-concentration properties (Definition 2.1) in Theorem 1.2. Moreover, this class of distributions contains strongly log-concave distributions, and assuming Kannan–Lovasz–Simonovits (KLS) conjecture, contains all log-concave distributions (Definition 2.2-2.6). Later in theorem 1.3, they design a universal tester learner for Halfspaces with Massart noise (the labels are flipped by an adversary with probability $\eta <1/2$).

Some previous and concurrent works for single distribution D^* assumption use approximate moment matching techniques, where the algorithm tests if the low moments of the input distribution D approximately matches with that of D^*. However it is not clear if this same approach can be applied for a collection of distributions. This has also been discussed in the introduction.

The authors first design testers for testing bounded disagreement (Lemma 3.1) and of testing anti-concentration properties (Lemma 3.2) which uses a tester for testing hypercontrctivity. It is known that any distribution $D$ with bounded Poincare constant is hypercontractive in Sum-of-Squares (SOS) framework. Thus the authors run a polynomial time semidefinite program for the later purpose. Next in Section 4, the authors design the final tester (Theorem 4.1). They use a surrogate loss minimization technique for the surrogate loss function defined in Equation 4.1. The main idea is that stationary points of surrogate loss are close to some optimal vector corresponding to the halfspace (the related lemma is Lemma 4.3). Lemma 4.3 contains bounds for both Massart and adversarial noise settings.

It would be interesting if tester-learners can be designed for function classes other than halfspaces. The authors also discusses this at the end of Section 1.


**Strengths:**

Overall this is a nice result which generalizes several previous works designed for a single marginal distribution assumption and studies for the setting of a collection of marginal distributions.

**Weaknesses:**

What is the usefulness of the tester-learner model in real life applications.

**Questions:**

None

**Limitations:**

It would be interesting if tester-learners can be designed for function classes other than halfspaces. The authors also discusses this at the end of Section 1.

---

### Decision · Program_Chairs · 2023-09-21

**Decision:**

Accept (oral)

**Comment:**

This is a strong result about a fundamental, well-motivated problem, requiring interesting and non-trivial ideas.